# Mapping brain structural differences and neuroreceptor correlates in Parkinson's disease visual hallucinations

Miriam Vignando [1✉], Dominic ffytche [2], Simon J. G. Lewis[3], Phil Hyu Lee[4], Seok Jong Chung[4], Rimona S. Weil[5,6], Michele T. Hu[7,8], Clare E. Mackay[7,9], Ludovica Griffanti [7,9], Delphine Pins[10], Kathy Dujardin [10], Renaud Jardri [10], John-Paul Taylor[11], Michael Firbank [11], Grainne McAlonan[12], Henry K. F. Mak [13], Shu Leong Ho[13] & Mitul A. Mehta [1]

Parkinson's psychosis (PDP) describes a spectrum of symptoms that may arise in Parkinson's disease (PD) including visual hallucinations (VH). Imaging studies investigating the neural correlates of PDP have been inconsistent in their findings, due to differences in study design and limitations of scale. Here we use empirical Bayes harmonisation to pool together structural imaging data from multiple research groups into a large-scale mega-analysis, allowing us to identify cortical regions and networks involved in VH and their relation to receptor binding. Differences of morphometrics analysed show a wider cortical involvement underlying VH than previously recognised, including primary visual cortex and surrounding regions, and the hippocampus, independent of its role in cognitive decline. Structural covariance analyses point to the involvement of the attentional control networks in PD-VH, while associations with receptor density maps suggest neurotransmitter loss may be linked to the cortical changes.

[1] Department of Neuroimaging, King's College London, Institute of Psychiatry, Psychology and Neuroscience, De Crespigny Park, London SE5 8AF, UK. [2] Department of Old Age Psychiatry, King's College London, Institute of Psychiatry, Psychology and Neuroscience, De Crespigny Park, London SE5 8AF, UK. [3] ForeFront Parkinson's Disease Research Clinic, Brain and Mind Centre, School of Medical Sciences, University of Sydney, Camperdown, NSW, Australia. [4] Yonsei University College of Medicine, Seoul, South Korea. [5] Dementia Research Centre, University College London, 8-11 Queen Square, London WC1M 3BG, UK. [6] Wellcome Centre for Neuroimaging, University College London, London, UK. [7] Oxford Parkinson's Disease Centre, Oxford, UK. [8] Nuffield Department of Clinical Neurosciences, University of Oxford, Oxford, UK. [9] Wellcome Centre for Integrative Neuroimaging, Oxford Centre for Human Brain Activity, Department of Psychiatry, University of Oxford, Oxford, UK. [10] Univ. Lille, Inserm, CHU Lille, U1172 - Centre Lille Neuroscience & Cognition, 59000 Lille, France. [11] Newcastle University, Translational and Clinical Research Institute, Biomedical Research Building, Campus for Ageing and Vitality, Newcastle Upon Tyne NE4 5PL, UK. [12] King's College London, Institute of Psychiatry, Psychology and Neuroscience, De Crespigny Park, London SE5 8AF, UK. [13] Division of Neurology, Dept of Medicine, LKS Faculty of Medicine, University of Hong Kong, Hong Kong, Hong Kong. ✉email: miriam.vignando@kcl.ac.uk

Parkinson's disease (PD) is a neurodegenerative disorder primarily characterised by motor symptoms, mainly related to the loss of neurons in the substantia nigra projecting to the basal ganglia[1]. Patients with Parkinson's disease (PD), in addition to the typical motor symptoms, commonly experience a variety of non-motor symptoms, including psychiatric ones[2]. Among these, visual hallucinations (VH) and related visual phenomena form a spectrum of symptoms referred to as Parkinson's disease psychosis[3] (PDP). There is a continuum of experiences typically characterising PDP with patients initially experiencing minor hallucinations (perception of presence or passage) and illusions that progress to formed hallucinations (initially with insight preserved); in rare cases, patients may also experience multimodal hallucinations and delusions[4]. Such symptoms may affect up to 70% of PD patients in more advanced stages of the illness[5] in the context of dopamine therapy but do not show a clear relationship between medication introduction or dose suggesting they are not simply medication side-effects[4]. VH predict a range of poor outcomes including more rapid cognitive decline and development of dementia[6–8] and increased likelihood of a move from independent living to a care home[9,10]. It is difficult to determine how VH might be related to these poor outcomes without a clear understanding of the brain systems involved in VH[4].

Imaging studies of VH in PD to date have been based on relatively small samples and have used differing designs that variously control for the degree of cognitive decline, stage of PD and dopamine medication. This makes it difficult to disentangle brain changes related specifically to VH mechanisms as distinct from those related to cognitive decline, PD stage or medication effects. As a result, a heterogeneous array of structural differences has been reported. Depending on whether or not cognition is controlled for, some studies have found volume reductions in specific regions that have not been replicated in other studies including: hippocampus[11], cerebellum[11,12], lateral, superior and medial frontal cortex[12–14], thalamus[15], and different subregions of visual association cortex, broadly defined to include the lateral occipital cortex, ventral occipito-temporal cortex (ventral stream), and visual parietal lobe (dorsal stream)[12,16,17].

A meta-analysis[18] utilising the previously reported regional differences demonstrated very little consistency across studies. It suggested this may be due to heterogeneity in structural brain correlates of VH, varying sensitivity to detect differences in multiple small studies, or the involvement at different locations of a unifying brain network whose dysfunction results in VH[18]. While meta-analytical techniques can be useful to collate findings from different studies and help understand the consistency of brain regions involved, there are limitations in their ability to include variables such as cognition, medication dose, PD stage, and duration as covariates, given that these are usually incorporated into the analyses at the study level and each study contributes a different set of regions to the meta-analysis. In contrast, mega-analyses bring together subject-level data across sites in one analysis, which presents a number of advantages. These include methodological rigour, with shared quality control and pre-processing pipelines, including software version control and the ability to include unpublished data or published data that was not used in the primary analysis (e.g., structural data collected for functional imaging studies). The same experimental design model and covariates can be applied uniformly across the data set helping address design variations in previous studies. Another advantage of the increased sample size is the additional power to explore morphometric features such as cortical thickness and cortical surface area along with undertaking complex analyses, such as structural covariance. Cortical thickness and surface area are considered as orthogonal components, which are affected by distinct underlying genetic processes[19] and can be considered separate morphometric components in ageing and disease[20,21]. The main correlate of cortical volume is cortical surface area, but volume loss is best captured by cortical thickness[21,22]. Separate measurement and analysis of these two components thus offer a better understanding of the underlying cortical changes associated with VH in PD than volume measures alone. Finally, mega-analyses create a valuable resource that can evolve and be made available to the wider neuroimaging community, especially important in PDP given that such patients are difficult to recruit and scan.

Several neurotransmitter systems have been associated with VH in PD. Initially, VHs were proposed to be a side effect of dopaminergic medication[23], but later evidence has led to a revision of this view. Current consensus is that dopaminergic medication interacts with disease-related susceptibility factors in PD to cause VH, rather than as a simple side effect[3]. Cholinergic pathways have also been implicated in VH[24,25], with neurodegeneration in brainstem and forebrain cholinergic nuclei[24] and electrophysiological measures of cholinergic function reduced in patients with VH[26]. Recently, a role for serotonergic dysfunction in VH has been suggested[27], linked to alterations in 5-HT2$_A$ receptor density[28,29] (for a review, see ref. [30]).

In this work, we perform a mega-analysis of PD with VH compared to PD without VH. This enables analyses that are not available to smaller scale studies to help explore the mechanisms of VH. Specifically, we are able to determine the regional cortical thickness and surface area changes associated with VH and relate these morphometric features to measures of symptom severity in a subgroup, where finer-grain clinical detail is available, finding widespread cortical thinning and in particular in regions previously associated with VH, such as the ventral visual stream. We perform a principal component analysis to identify smaller-scale morphometric differences within a high dimensional set of regions. The aim is to identify unifying dimensions across the multiple regions different in PD with VH compared to PD without VH in order to have a better understanding of which of the models proposed thus far may account for such symptoms. We find two separate components, a frontal and an occipital one. In addition, we perform an exploratory structural network analysis to highlight associations between regions and clusters of connections linked to VH showing an involvement of the attentional control networks in PD-VH. Structural covariance allows us to assay covariation of differences in grey matter morphology between different brain structures, providing information on which regions similarly change in thickness or surface area. In order to understand the neurochemical associations of these changes, we also test the hypothesis that structural differences are related to the spatial variation in subtypes of receptors for which high resolution PET atlases are available (dopamine and serotonin) with results suggesting neurotransmitter loss may be linked to the cortical changes.

## Results

**Patient characteristics.** The final dataset consisted of 493 participants (193 F), of which 135 were PD-VH. Each individual study had matched their participants for age, gender, disease onset, MMSE, UPDRS-III and levodopa equivalent daily dose (LED), except for two studies (unpublished data) where MMSE score was lower in PD-VH, two studies (selection of published and unpublished data) with UPDRS-III scores higher in PD-VH, and two studies (one unpublished data) where gender was not matched (Table 1). We included the unpublished data in separate within groups ANOVAs to check on which variables the groups differed (Table 1). In order to understand potential biases in the larger,

**Table 1 Demographics and clinical information by group.**

| Study | N patients | Gender | Age | Onset | MMSE | UPDRS III | LED |
|---|---|---|---|---|---|---|---|
| Shin et al.[15] | 41 PD-VH<br>56 PD-noVH | PD-VH 20F<br>PD-noVH 30F<br>$\chi^2 = .64$,<br>$p = 0.6$ | 71.8 ± 6.1<br>71.1 ± 5.9<br>$p = 0.5$ | 3.7 ± 3.3<br>2.7 ± 3.0<br>$p = 0.1$ | 25.3 ± 3.0<br>25.7 ± 2.9<br>$p = 0.4$ | 24.1 ± 10.4<br>21.6 ± 11.0<br>$p = 0.09$ | 486.4 ± 303.1<br>386.3 ± 241.7<br>$p = 0.07$ |
| Shine et al.[51] + unpublished data | 30 PD-VH<br>58 PD-noVH | PD-VH 12F<br>PD-noVH 12F<br>$\chi^2 = .37$,<br>$p = 0.05$ | 66.6 ± 7.2<br>66.4 ± 8.6<br>$p = 0.9$ | 6.0 ± 3.9<br>5.4 ± 3.5<br>$p = 0.5$ | 28.8 ± 1.6<br>29.4 ± 1.3<br>$p = 0.04$ | 32.0 ± 13.4<br>27.8 ± 13.4<br>$p = 0.2$ | 664.3 ± 495.2<br>706.8 ± 502.7<br>$p = 0.7$ |
| Firbank et al.[48] (only non-dementia data retained) | 10 PD-VH<br>11 PD-noVH | PD-VH 2F, PD-noVH 2F<br>$\chi^2 = .92$,<br>$p = 0.7$ | 75.0 ± 3<br>71.7 ± 5.3<br>$p = 0.2$ | 10.2 ± 8.2<br>10.1 ± 7.6<br>$p = 0.9$ | 25.9 ± 1.6<br>27.2 ± 2.4<br>$p = 0.2$ | 51.7 ± 22.2<br>30.50 ± 14.73<br>$p = 0.05$ | 469.9 ± 31.3<br>693.4 ± 411.2<br>$p = 0.2$ |
| Yao et al.[46] (part of) | 6 PD-VH, 21 PD-noVH | PD-VH 1F<br>PD-noVH 10F<br>$\chi^2 = .17$,<br>$p = 0.2$ | a<br>64.2 ± 5.6<br>62.5 ± 7.2<br>$p = 0.6$ | a<br>10.0 ± 3.5<br>8.4 ± 5.1<br>$p = 0.4$ | a<br>27.6 ± 2.4<br>28.5 ± 1.7<br>$p = 0.09$ | a<br>20.9 ± 10.6<br>18.0 ± 12.9<br>$p = 0.5$ | a<br>978.7 ± 361.3<br>704.9 ± 519.4<br>$p = 0.2$ |
| Lefebvre et al.[55] unpublished structural data | 16 PD-VH<br>15 PD-noVH | PD-VH 5F<br>PD-noVH 4F<br>$\chi^2 = .78$,<br>$p = 0.8$ | 63.8 ± 6.2<br>63.1 ± 3.9<br>$p = 0.7$ | 8.7 ± 4.47<br>7.3 ± 5.25<br>$p = 0.4$ | 27.9 ± 1.3<br>28.9 ± 1.2<br>$p = 0.02$ | 26.3 ± 7.8<br>21.4 ± 8.1<br>$p = 0.1$ | 871.2 ± 406.7<br>765.9 ± 263.6<br>$p = 0.4$ |
| Lawn and ffytche[47] | 7 PD-VH<br>9 PD-noVH | PD-VH 4F<br>PD-noVH 3F<br>$\chi^2 = 15$,<br>$p = 0.3$ | 68.7 ± 7.2<br>66.1 ± 6.5<br>$p = 0.5$ | 8.1 ± 5.4<br>5.8 ± 4.1<br>$p = 0.3$ | 26.8 ± 4.<br>29.7 ± 0.5<br>$p = 0.06$ | 40 ± 13.4<br>25.6 ± 6.6<br>$p = 0.01$ | 806.6 ± 464.5<br>709.6 ± 530.7<br>$p = 0.7$ |
| Oxford Discovery Cohort, unpublished[b] (Baig et al.[57], Griffanti et al.[58]) | 7 PD-VH<br>103 PD-noVH | PD-VH 5F<br>PD-noVH 36F<br>$\chi^2 = 3.7$,<br>$p = 0.05$ | 60.35 ± 10.42<br>63.86 ± 10.35<br>$p = 0.4$ | 2.4 ± 1.6<br>2.0 ± 1.0<br>$p = 0.6$ | 28.7 ± 1.4<br>28.6 ± 13<br>$p = 0.8$ | 23.0 ± 12.7<br>23.8 ± 10.3<br>$p = 0.8$ | 414.6 ± 220.1<br>321.4 ± 242.9<br>$p = 0.3$ |
| Zarkali et al.[56] for demographics (T1 data submitted) | 18 PD-VH<br>85 PD-noVH | PD-VH 13F<br>PD-noVH 35F<br>$\chi^2 = 5.7$,<br>$p = 0.02$ | 64.3 ± 8.3<br>64.1 ± 7.7<br>$p = 0.9$ | 4.2 ± 2.4<br>4.1 ± 2.5<br>$p = 0.8$ | 29.0 ± 1.6<br>28.9 ± 1.1<br>$p = 0.9$ | 21.7 ± 11.0<br>24.1 ± 13.1<br>$p = 0.4$ | 461.5 ± 269.2<br>415.6 ± 162.5<br>$p = 0.5$ |

Each row represents the data present in the study for each group. Not all groups could share raw clinical data. In those cases, we reported the information of the original publication to show that in the original study there was no difference within groups in terms of PD and medication. Within the different groups, gender was not matched for the UCL and Sydney samples. MMSE was not matched in the Sydney sample, and UPDRS-III was not matched for the KCL (Prof. ffytche) sample and the Newcastle sample. The values reported are means, standard deviations and the $p$ values are the result of one-way ANOVAs within each group. We performed a meta-analysis including raw values when possible (and excluding missing). PD-VH had lower MMSE score (difference is 1 point (sd 0.1; with 29.4 for noVH and 28.8 for VH) and higher UPDRS-III score (difference is 5.8 (sd 2.7) with both groups being within the "moderate" category as reported in Martínez-Martín et al.[96]). Thus, for our sensitivity analysis in the $N = 440$ and in the NPI subsample, we used all of these relevant variables and their interactions as covariates (see 2.3 and Supplemental Information S3b and S4).
[a]For this variable we did not have raw data and we report the values provided in the original paper.
[b]Non-motor symptoms[56], T1 data[57] separately published, but not in a publication studying them together.

combined sample we also included the data in meta-analyses (Supplementary Note 2) and in an ANOVA including the whole mega-analysis sample. While the ANOVAs and the meta-analysis demonstrated groups do not differ on age, gender, disease onset and medication (levodopa equivalent dose), the mega-analysis ANOVA shows that there is a difference of 2.19 years in age [$F(1,491) = 6.56$, $p = 0.01$] (PD-VH = 67.85, SD = 7.74; 62 F, and PD-noVH = 65.66, SD = 8.71; 131 F). There is also a greater proportion of females in the PD-VH group ($\chi^2 = 3.585$, $p = 0.06$). The meta-analysis also showed a difference in MMSE and UPDRS-III (see Table 1 for means and SD and S2 for details). Thus, for a smaller subsample of patients ($N = 146$) for which we had complete clinical information with patients not differing on clinical variables we carried out a sensitivity analysis (see 2.3 and S4) and we also carried out a sensitivity analysis on a $N = 440$ subsample for which all relevant variables were available (see 2.3 and S3b).

Morphometrics were harmonised (Supplementary Note 1) and we did not find significant differences in total intracranial volume (TIV) [$F(1,493) = 0.043$, $p = 0.84$] or total brain volume [$F(1,493) = 2.488$, $p = 0.115$], but in total grey matter volume [$F(1,493) = 5.41$, $p = 0.02$] (see Supplementary Note 2).

**Patients with hallucinations (PD-VH) vs. without (PD-noVH) multivariate analysis of variance.** We initially ran an exploratory one-way ANOVA to perform an initial feature selection for the

MANCOVA models. Significant results were corrected for multiple comparisons and only regions surviving this phase were entered in the final models.

*Cortical thickness.* Pairwise comparisons on between-subjects effect show that lower thickness in PD-VH was present in a widespread set of regions (see Fig. 1 depicting regions sorted by effect size and Supplementary Note 3 for tables and further details). No regions showed greater cortical thickness in PD-VH. A main effect of age [$F(88,401) = 4.26$, $\eta^2 = 0.48$, $p < 0.001$], gender [$F(88,401) = 1.83$, $\eta^2 = 0.29$, $p < 0.001$] and TIV [$F(88,401) = 2.66$, $\eta^2 = 0.37$, $p < 0.001$] was observed.

*Surface area.* We found reduced area in PD-VH in the right occipitotemporal gyrus/medial occipital cortex (see Fig. 1) (for all table and details see Supplementary Note 3a). A significant main effect of age [$F(1,492) = 33.7$, $\eta^2 = 0.65$, $p < 0.001$] and TIV [$F(1,492) = 70.29$, $\eta^2 = 0.13$, $p < 0.001$] was observed.

*Subcortical volumes.* We found a lower volume for PD-VH in the left cerebellar white matter and in the bilateral amygdala (see Supplementary Note 3a). A significant main effect of age [$F(16,473) = 14.58$, $\eta^2 = 0.34$, $p < 0.001$], gender [$F(16,473) = 2.06$, $\eta^2 = 0.07$, $p = 0.03$] and TIV [$F(16,473) = 31.27$, $\eta^2 = 0.53$, $p < 0.001$] was observed.

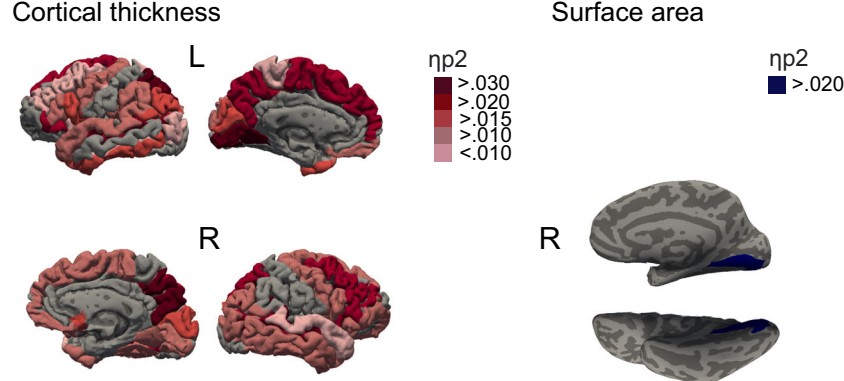

### a) Main sample (N=493) PD-VH < PD-noVH

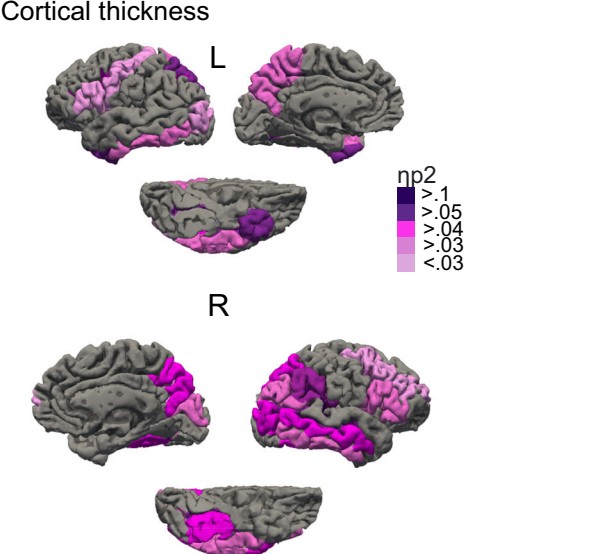

### b) Confirmatory subgroup analysis (N=146) PD-VH < PD-noVH

**Fig. 1 Group differences for PD-VH (patients with hallucinations) vs. PD-noVH (without hallucinations). a** The figure is a visual summary of the results for the multivariate between subjects (VH/noVH) ANCOVA, using age, gender and TIV as covariates. Shown are regions whereby PD-VH had decreased cortical thickness (right) and surface area (left). Widespread decreased thickness was found in PD-VH; the regions with the greatest effect size were occipitotemporal (occipitotemporal and occipital sulci, inf. temporal gyrus), parietal (precuneus, intraparietal sulcus) and frontal (superior and middle frontal gyri, inferior frontal gyrus) regions. Surface area was reduced in PD-VH in the right occipitotemporal gyrus/medial occipital cortex. **b** The figure is a visual summary of the results for the multivariate between subjects (VH/noVH) ANCOVA, using age, gender, TIV, LED, disease onset, UPDRS-III and MMSE as covariates for the replication cohort ($N = 146$, NPI subsample) (for details see 2.3 and S4). Regions are colour coded by effect size (details in S3). Results were corrected for multiple comparisons (FDR) and pairwise comparisons (Bonferroni) (see "Methods" section). Individual statistical values for each region are provided in the Supplemental Information S3 in tables, one for each morphometric.

Finally, the results were supported by a sensitivity analysis with all groups minus one (for all groups; see Supplementary Note 3c for details).

**Subgroup analysis**. We carried out two subgroup sensitivity analyses. For a subsample of 440 individuals (121 PD-VH) we carried out the same models as in the previous paragraph but adding as covariates LED, onset, MMSE and UPDRS-III, besides age, gender and TIV. Results were consistent with those presented on the full sample for thickness, area and volume (see Supplementary Note 3b for details).

For the subsample for which we have Neuropsychiatric Inventory (NPI) hallucinations subscale scores (frequency * severity), focussing on VH, we ran a sensitivity analysis and correlational analyses to explore the relationship of VH severity and cortical thickness. The NPI sample consists of 146 participants

(67 PD-VH, 79 PD-noVH). In the individual studies making up this dataset participants do not differ for age, gender, TIV, medication, cognition, onset, and PD severity (UPDRS-III). We compared PD-VH and PD-noVH on these variables with a one-way ANOVA finding that participants do not differ on age, gender, TIV, medication, cognition, onset and MMSE score. For one of the studies we had >20 missing values for UPDRS-III and decided to fill those values with the sample mean to use UPDRS-III as a covariate (but see Supplementary Note 4 for detailed comparisons). We carried out multivariate MANCOVAs to replicate the results from the main group, adding disease onset, medication, cognition and PD severity as additional covariates, and, when necessary, interactions between inter-correlated covariates were added to the model. No significant main effect of age, MMSE, LED, UPDRS, onset, age * LED + age * MMSE + onset * LED + onset * UPDRS-III + LED * MMSE was observed, but only of gender

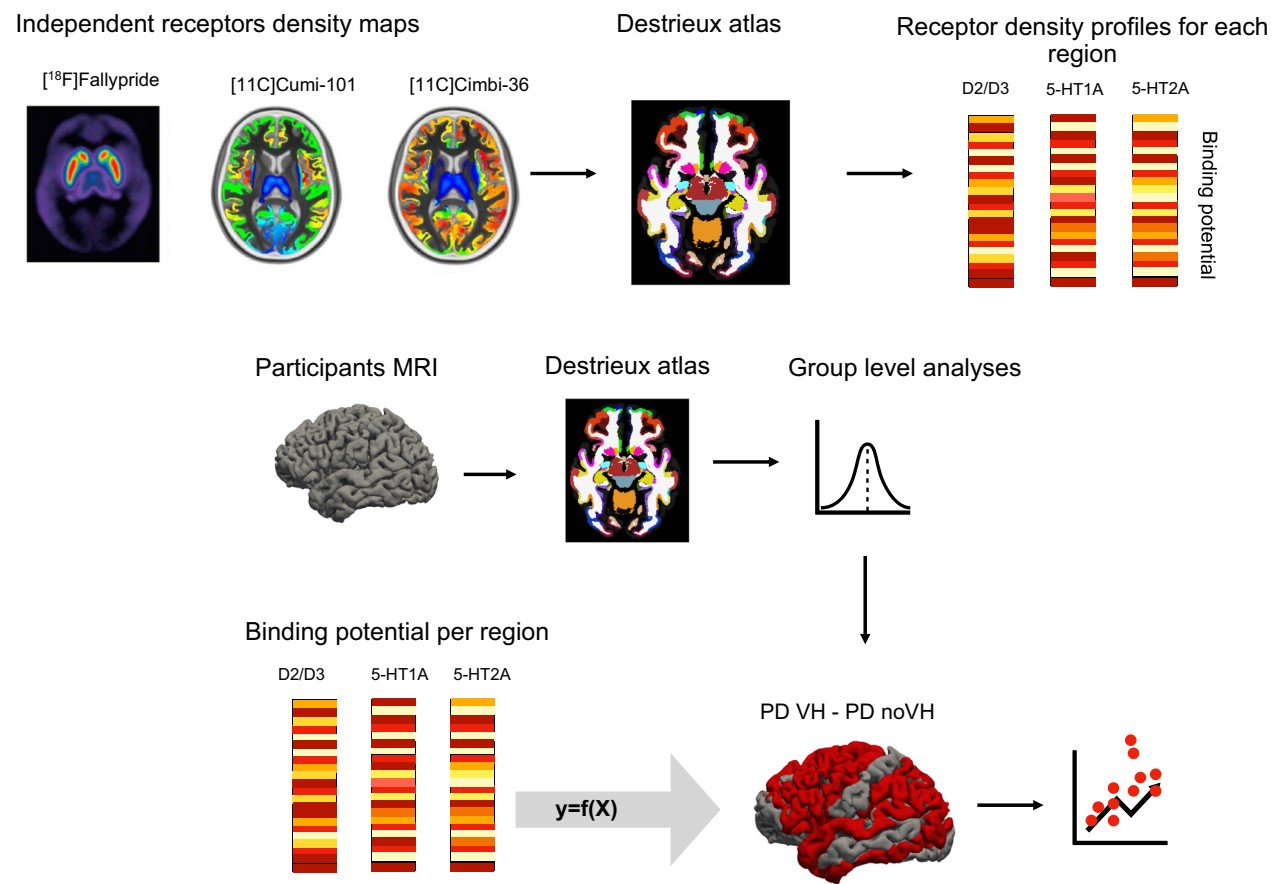

**Fig. 2 Receptors density profiles: methods.** Procedure and rationale of the regression models. Both the independent receptor density maps and our participants' MRI scans were parcellated with the Destrieux atlas. Cortical thickness values were extracted for each region of the atlas for the participants' scans, and binding potential was extracted for each region of the atlas for the receptor density maps. Each receptor's binding potential was used in separate models as a predictor of difference of the means of thickness between PD-VH (patients with hallucinations) vs. PD-noVH (without hallucinations).

and TIV, as in the full sample models. Results were consistent with those found for the main sample (see Fig. 1b and Supplementary Note 4 for details).

When correlating the NPI score with morphometrics in PD-VH only ($N = 67$) and controlling for age, LED, MMSE, and UPDRS-III, inverse correlations were significant for right hemisphere cortical thickness in the intraparietal sulcus ($r = −0.24$, $p = 0.05$), the superior temporal sulcus ($r = −0.26$, $p = 0.04$), the Jensen sulcus (between the anterior and posterior rami of the IPS) ($r = −0.28$, $p = 0.02$) and the cingulum (marginalis) ($r = −0.298$, $p = 0.017$), and the right postventral cingulum ($r = −0.27$, $p = 0.03$). In addition, we checked possible correlations between NPI and confounding variables, finding that none of such correlations was significant (UPDRS-III * NPI: $r = −0.012$, $p = 0.92$, age* NPI: $r = −0.09$, $p = 0.47$, NPI*onset: $r = −0.05$, $p = 0.68$, NPI* LED: $r = −0.012$, $p = 0.93$, NPI* MMSE: $r = 0.18$, $p = 0.14$).

**Receptors density maps regression models.** After parcellating the receptor densities maps of D2/D3, 5-HT$_{2A}$ and 5-HT$_{1A}$ receptors (derived from independent healthy participants, see "Methods" section) using the Destrieux atlas to ensure that density and morphometric data were aligned, we explored the relationship between the differences in cortical thickness between PD-VH and PD-noVH (see Figs. 2 and 3). Separate linear models were carried out for each receptor density map, with separate

outlier detection processes (Cook's distance). These correlations were assessed both with a model including the morphometric difference values only of regions where we found a significant difference, and with a model including the morphometric difference values in all regions. We also performed correlational analyses taking into account spatial autocorrelation to correct for the spatial similarity of neighbouring regions. The maps used were independent atlases built on healthy subjects' PET data (see "Methods" section and Fig. 2).

*Cortical thickness.* The model with 5-HT$_{2A}$ binding potential as predictor and the mean difference of cortical thickness (PD-VH - PD-noVH) as dependent variable was significant for the subset of regions, where the groups were shown to differ in the main MANCOVA ($\beta = −0.35$, $t = −2.8$, $p = 0.006$, when corrected for spatial autocorrelation on a symmetrical subset of regions (see methods for details) $p = 0.01$; confidence intervals estimated with bootstrapping $−1122.7$, $−278.7$), whereas no relationship was observed when considering all the atlas regions ($\beta = −0.03$, $t = −0.39$, $p = 0.7$, $p = 0.36$ corrected for spatial autocorrelation). A similar result was observed for 5-HT$_{1A}$ (significant regions: $\beta = −0.32$, $t = −2.5$, $p = 0.01$, confidence intervals estimated with bootstrapping $0.00$, $0.00$; corrected for spatial autocorrelation $p = 0.003$; all regions: $\beta = −0.02$, $t = −0.21$, $p = 0.8$, $p = 0.26$ corrected for spatial autocorrelation) and for D2/D3 receptor for

## Receptors density profiles: cortical thickness results

### a) Regions shown to differ in the main MANCOVA

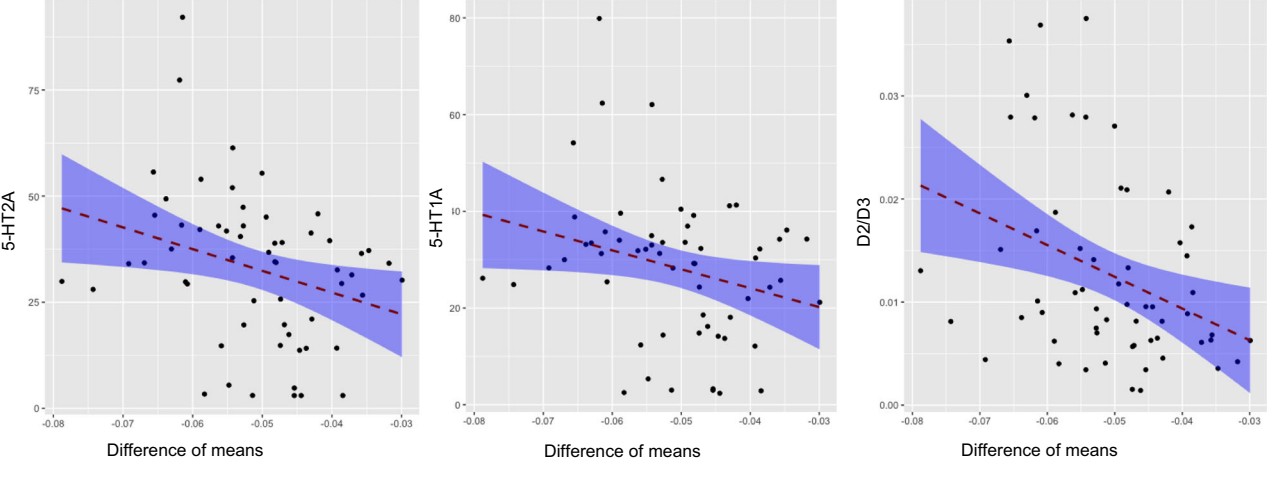

### b) All regions

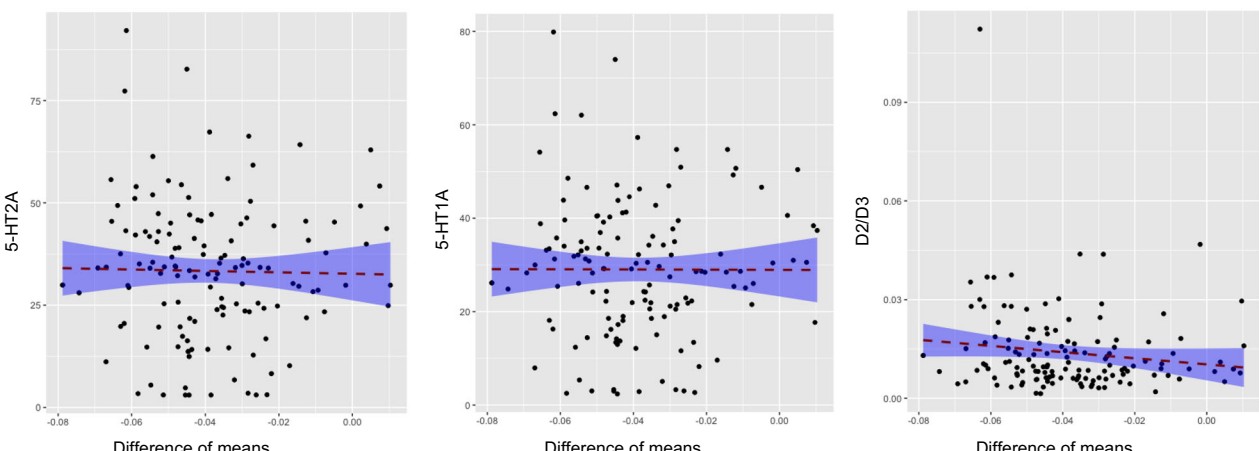

**Fig. 3 Receptors density profiles: Results of regression models.** Shown are the results of the linear regression (two-sided) models with receptor density as predictor and difference of the means (with hallucinations–without) of cortical thickness as dependent variable. **a** Regions which were different between groups as dependent variable and **b** the models for all regions. Results are reported for 5-HT$_{2A}$, 5-HT$_{1A}$, and D2/D3 receptors binding potential and cortical thickness, further details are in S6. In addition, the *p* value of the correlation between thickness and each receptor map taking into account spatial autocorrelation is reported in the text. The shaded area represents the confidence interval.

the model with difference regions (significant regions: $\beta = -0.41$, $t = -3.3$, $p = 0.002$, confidence intervals estimated with bootstrapping $-0.65$, $-0.24$; corrected for spatial autocorrelation $p = 0.005$; all regions: $\beta = -0.15$, $t = -0.1.7$, $p = 0.08$, $p = 0.002$ corrected for spatial autocorrelation) (see Fig. 3 for results). When repeating the models including only the regions where no difference was observed, 5-HT$_{1A}$ and 5-HT$_{2A}$ were not significant predictors, whereas D2/D3 was a significant predictor (for details see Supplementary Note 6). In addition, we compared the slopes of the models with the differing regions, finding no significant difference between models with different receptors. We did not compare the slopes for analyses with all regions as D2/D3 was the only significant predictor; the same applies to the models with the regions that showed no difference in the MANCOVA (see Supplementary Note 6 for further details).

The same model was carried out also for subcortical volumes yielding significant results for all receptors (see Supplementary Note 6).

**Principal components analysis (PCA).** We used PCA to reduce the dimensionality of the dataset while preserving variability. This technique was selected to identify underlying clusters to clarify the results from the widespread pattern of cortical thinning described in the group-level analysis. PCA allows us to identify latent patterns, summarising the brain regions that contribute the most to such patterns in dimensions by identifying the directions that have the widest variations, allowing for more precise inference about the data.

*Cortical thickness.* The PCA returned two dimensions with eigenvalues >1 explaining 76.27% of total variance. The principal components for Dimension 1 (eigenvalue = 3.24, 53.98% of variance) were the bilateral superior frontal gyrus and the right middle frontal gyrus. These were the regions with the highest cosine squared index. The PC for Dimension 2 (eigenvalue = 1.34, 22.29% of variance) were the bilateral cuneus, and the right occipitotemporal gyrus (see Fig. 4, for scree plots and regional contribution plots see Supplementary Note 7 and Supplementary Fig. 4).

## PCA dimensions for cortical thickness

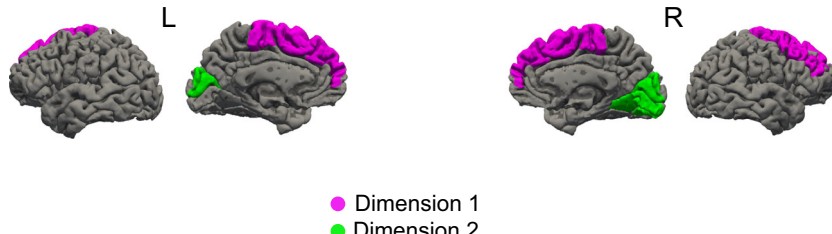

● Dimension 1
● Dimension 2

**Fig. 4 Graphical representation of the regions contribution to each of the Dimensions resulting from the principal component analysis.** Regions contributing to dimension 1 and 2, cortical thickness. Dimension 1 (pink): superior frontal gyri, the right middle frontal gyrus. Dimension 2 (green): bilateral cuneus and right occipital superior gyrus.

**Structural covariance analysis**. To explore and characterise the grey matter network-level organisation of PD-VH and PD-noVH patients for cortical thickness and surface area we carried out structural covariance analyses, that assess the covariation of differences in grey matter morphology between different brain structures. This sample includes 467 individuals 118 PD-VH (56 F), mean age = 67.19, sd = 7.77, 349 PD-noVH (129 F) mean age = 65.7, sd = 8.77, not differing on age. Age and gender were used as covariates. After specifying a general linear model for each region, the structural covariance matrices (68 × 68) of the two groups were defined by estimating the inter-regional correlation between model residuals of thickness and area (in separate models).

For cortical thickness, the cell-by-cell comparisons of residuals' inter-regional correlation coefficients (carried out with *cocor* and FDR corrected) highlighted differences in the interregional covariance (see Fig. 5 for a visual representation). Among the regions shown, the left PGH showed particularly high covariance with several other regions, among which but not limited to, the regions shown in the figure. Overall, inter-regional correlations were greater for the PD-VH group (see Supplementary Note 8).

Hubs, that is nodes (here regions) that are thought to strongly contribute to the global network function, were identified in frontal, parietal and occipital regions for the PD-noVH group, and in frontal, temporal and parietal regions for the PD-VH group (Fig. 6a). Permutation tests for vertex-level measures returned differences in betweenness centrality, which was greater in PD-VH in the left and right lingual gyrus, in the left lateral occipital gyrus and the right SPL (p FDR < 0.05). Communities are sets of brain regions characterised by denser and stronger relations among themselves, if compared with regions of other communities. Structural covariance-based communities have been found to replicate neighbourhoods observed with seed-based approaches in fMRI and DTI (see "Methods" section for details). The first community in the PD-VH group comprised mainly occipitotemporal regions, with the second involving parietal and some frontal regions. In the PD-noVH group, the first community consisted of mostly frontoparietal regions whereas the second comprised occipitoparietal regions (Fig. 6b). In addition, the PD-noVH group showed higher modularity, as assessed with bootstrapping (mean = 0.29 SD = 0.02, CI 0.25, 0.36 at density 0.13) (for communities by lobe, see Supplementary Note 8).

Finally, we found no significant correlation between difference of the means in thickness and difference of the means in graph-level measures of interest.

For surface area significant (FDR corrected) differences in interregional covariance were observed bilaterally in the rostral MFG, fusiform gyrus, and IPL; in the left caudal MFG, lateral occipital gyrus, SPL, and insula and in the right anterior and

posterior cingulate, and IFG pars opercularis, with some similarities to what observed for thickness (see Fig. 7 for a visual representation and Supplementary Note 8). Among these, the left caudal MFG, left IPL, left LOG, left paracentral gyrus, left temporoparietal, left insula, and right IPL, right paracentral, right IFG opercularis, right precuneus showed particularly high covariance with several other regions.

Hubs were identified mainly in occipitotemporal and frontal regions for the PD-noVH group and in frontal, temporal and occipital regions for the PD-VH group, also found in the PCA (Fig. 8). In accordance with this result, vertex-level permutation tests returned differences in betweenness centrality the left fusiform gyrus; in addition, differences ($p_{FDR} < 0.05$) were observed for the middle orbitofrontal gyrus, IFG orbitalis and triangularis, whereby centrality was greater for PD-noVH in these regions, but greater for PD-VH in the left caudal MFG and in the right SFG. The first community in the PD-VH group is characterised by occipitotemporal and frontal and the second community by occipito-parietal and parietal regions only (Fig. 8b; representation by lobe is in Supplementary Note 8). In addition, PD-noVH showed greater modularity, as assessed with bootstrapping (0.29, CI 0.21, 0.36 density 0.13).

Finally, we explored possible correlations between graph metrics of interest (local and global efficiency) and difference in surface area, not only in the covariance sample, but also in the full sample and in the NPI sample, to further test the robustness of the findings. We found a significant positive correlation between difference of the means for surface area in the NPI subsample and the difference in local efficiency ($r = 0.345$, $p = 0.004$, $p = 0.008$ corrected for multiple comparisons), in the full sample ($r = 0.354$, $p = 0.003$, $p = 0.007$ corrected for multiple comparisons) and in the covariance sample ($r = 0.350$, $p = 0.003$, $p = 0.007$ corrected for multiple comparisons), whereby the greater the difference in the surface area, the greater the difference in the local efficiency. The regions with both the greatest area differences and efficiency differences were the bilateral lingual gyrus, lateral occipital gyrus, right cuneus and right insula. We thus ran regression models with local efficiency as a predictor and difference in area in each sample, using bootstrapping (10,000 cycles) to estimate the confidence intervals. We found that local efficiency is indeed a predictor of differences in surface area in the NPI subsample ($\beta = 0.34$, $t = 2.9$, $p = 0.02$, 11.84, 86.96), in the full sample ($\beta = 0.35$, $t = 3.1$ $p < 0.001$, 23.39, 76.67) and in the covariance sample ($\beta = 0.35$, $t = 0.3.1$, $p = 0.004$, 26.95, 79.38).

## Discussion

We have presented a mega-analysis of patients with Parkinson's disease with and without visual hallucinations, demonstrating

## PD-VH > PD-noVH inter-regional correlations: cortical thickness

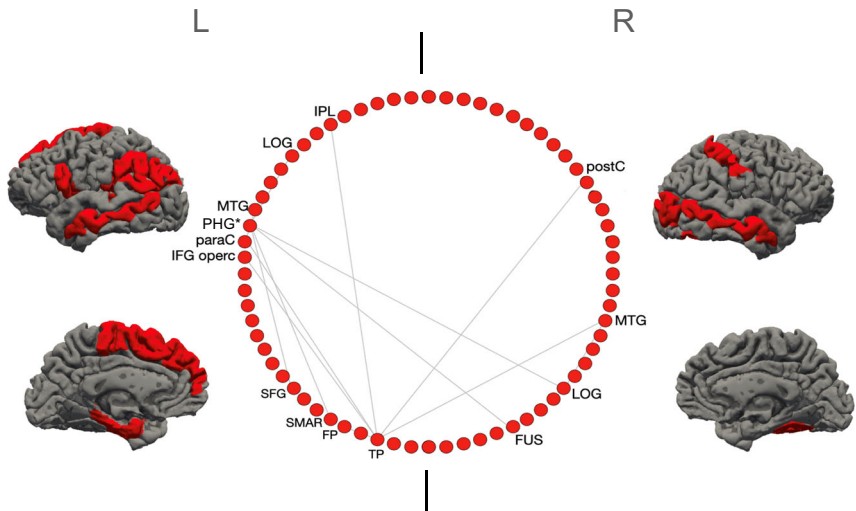

**Fig. 5 Regions with the most significant difference in inter-regional correlations of cortical thickness between groups: the inter-regional correlations for these regions were significantly greater for VH patients.** Shown in the circular plot, only the inter-regional correlations with a difference greater than 0.3 in the $r^2$; the region with the greatest number of significant inter-regional correlations, significant after multiple comparisons correction, is marked with a * (see also S8). IPL inferior parietal lobule, LOG lat. Occipital gyrus; MTG middle temporal gyrus, PHG parahippocampal gyrus; paraC paracentral gyrus; IFG opercularis inferior frontal gyrus, SFG superior frontal gyrus, SMAR supramarginal gyrus, FP frontoparietal thickness; TP temporoparietal, FUS fusiform gyrus, postC postcentral gyrus. The two vertical lines separate L and R hemisphere regions (left on left).

widespread alterations in brain structure, with differential effects for cortical thickness and surface area and examined their relationship to receptor distributions and network-level effects. Below we discuss the implications of the findings and their relationship to current theories of VH.

Cortical thickness and surface area are considered two separate components in ageing and disease[20,21,31] reflecting different aspects of the neurodegenerative process. Thickness loss relates to cortical layering and, by inference, cytoarchitecture, while surface area relates to gyral anatomy and, by inference, underlying white matter. Widespread reductions in cortical thickness in patients with hallucinations were identified in the occipital, parietal, temporal, frontal and limbic lobes. The regions of reduced thickness encompassed all cortical regions identified in previous structural imaging studies (for a review, see ref. [32]), suggesting previous variability may relate to stochastic effects introduced by relatively smaller samples and design differences. Interestingly, there is substantial overlap between the regions identified here and a functional network recently found involved in presence hallucinations in both healthy and PD participants[33]. With the larger sample of the mega-analysis, the extent of cortical regions involved appears wider than previously suspected. However, not all regions are equally affected and, notably, there appears to be a posterior asymmetry with relative sparing of the left ventral visual stream (ventral occipito-temporal cortex) compared to the homologous region in the right hemisphere. This region plays a key role in all models of VH in PD but a greater involvement of the right hemisphere has not been noted previously. The PCA analysis helped define key sub-regions within the extensive areas of cortical thinning that contributed most to the group difference, identifying a frontal and an occipital dimension. Of these, the cuneus and superior frontal gyrus bilaterally emerged as the dominant components. These regions have been reported in previous studies but do not play a prominent role in accounts of VH in PD. The cuneus is one of the earliest regions to show cortical atrophy in PDP, present at the earliest stages when only minor hallucinations occur[34], while cortical thinning in the

dorso-medial superior frontal gyrus has been reported in patients, months to years prior to the development of VH[35]. It may be that the prominence of these regions in the mega-analysis relates to the longer duration of these changes compared to other brain regions resulting in a greater consistency of thickness reduction between patients.

For surface area, the difference between groups was circumscribed with a reduction in the right occipitotemporal mediolingual gyrus for patients with VH. Taken together with the cortical thickness results, these extensive structural changes have been identified in the primary visual cortex and its surrounds in PD patients with VH and helps account for wide-ranging low-level visual deficits found (for a review, see ref. [36]). The result from the surface area analysis may imply additional gyral atrophy, sulcal widening and a reduction of underlying white matter.

The mega-analysis also allowed us to move beyond a binary comparison of VH versus noVH to examine brain regions linked to VH severity as measured by the NPI hallucination subscale score (a composite score derived from the product of frequency and distress ratings) and taking into account any variability associated with age, gender, TIV, medication, cognition, disease onset, and PD severity. Regions with reduced thickness for higher severity scores (negative correlation) were found in posterior parietal, posterior cingulate, and superior temporal cortex. Previous studies have associated these regions with mental rotation and visuospatial transformation[37] and imagery[38] for the IPS, and biological motion detection[39] for the STS. These processes are altered in patients with PD and VH[39,40], thus one can infer an involvement of these processes and these regions in VH severity. In addition, the IPS is also part of the dorsal attentional network, previously implicated in VH in PD[41] (see discussion below). These regions were also identified as hubs in the structural covariance analysis, discussed further below. As separate distress and frequency scores were available only for a part of the subsample, we were unable to analyse the two components of the severity score separately, thus we cannot disentangle whether these correlations are driven primarily by the frequency or distress

## a) Cortical thickness: Hubs

| Group | Hub Region |
|---|---|
|  |  |
| PD noVH | L Superior frontal gy.* |
|  | R Superior frontal gy. |
|  | L Supramarginal gy. |
|  | R Supramarginal gy. |
|  | L Inf. Parietal lobule |
|  | R Inf. Parietal lobule* |
|  | L Sup. Parietal lobule |
|  | R Sup. Parietal lobule |
|  | R Cuneus |
|  | R Lat. Occipital gy. |
|  |  |
| PD VH | L Superior frontal gy.* |
|  | R r Middle Frontal gy. |
|  | L r Middle Frontal gy. |
|  | L Sup. Temporal gy. |
|  | R Inf. Parietal Lobe* |

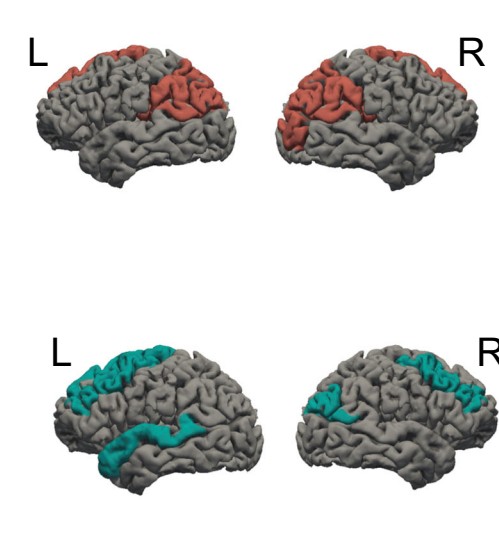

## b) Communities

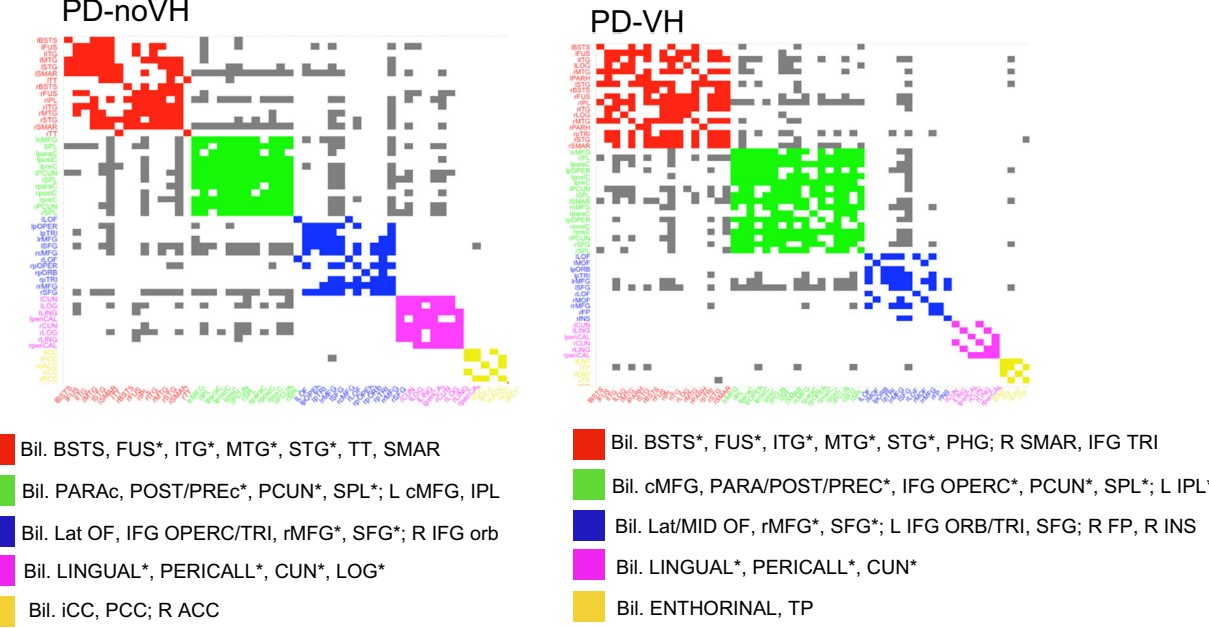

**PD-noVH**

- 🟥 Bil. BSTS, FUS*, ITG*, MTG*, STG*, TT, SMAR
- 🟩 Bil. PARAc, POST/PREc*, PCUN*, SPL*; L cMFG, IPL
- 🟦 Bil. Lat OF, IFG OPERC/TRI, rMFG*, SFG*; R IFG orb
- 🟪 Bil. LINGUAL*, PERICALL*, CUN*, LOG*
- 🟨 Bil. iCC, PCC; R ACC

**PD-VH**

- 🟥 Bil. BSTS*, FUS*, ITG*, MTG*, STG*, PHG; R SMAR, IFG TRI
- 🟩 Bil. cMFG, PARA/POST/PREC*, IFG OPERC*, PCUN*, SPL*; L IPL*
- 🟦 Bil. Lat/MID OF, rMFG*, SFG*; L IFG ORB/TRI, SFG; R FP, R INS
- 🟪 Bil. LINGUAL*, PERICALL*, CUN*
- 🟨 Bil. ENTHORINAL, TP

**Fig. 6 Hubs and communities: cortical thickness. a** Hubs identified based on efficiency, betweenness centrality and degree. Regions marked with a * are common hubs for both VH and noVH. **b** Communities identified for each group. Legend: red = 1st community, green = 2nd, blue = 3rd, pink = 4th, yellow = 5th. Only the first five communities are represented as they are the most informative ones. The regions identified for that same community also in the surface area analysis are marked with a *. BSTS = banks superior temporal sulcus, IPL inferior parietal lobule, SPL superior parietal lobule, LOG lat. Occipital gyrus, MTG middle temporal gyrus, STG superior temporal gyrus, cMFG caudal middle frontal gyrus, PHG parahippocampal gyrus, paraC paracentral gyrus, preC precentral gyrus, postC postcentral gyrus, IFG inferior frontal gyrus, SFG superior frontal gyrus, SMAR supramarginal gyrus, FP frontoparietal thickness, TT temporal transverse, FUS fusiform gyrus, CUN cuneus.

measures. However, our findings suggest a link between cortical structural changes and phenomenological aspects of VH severity as reflected in the combined measure.

In addition to the detailed analysis of the cerebral cortex we were able to examine the volumes of subcortical structures as well. Bilateral volume reduction was found in the amygdalae in the

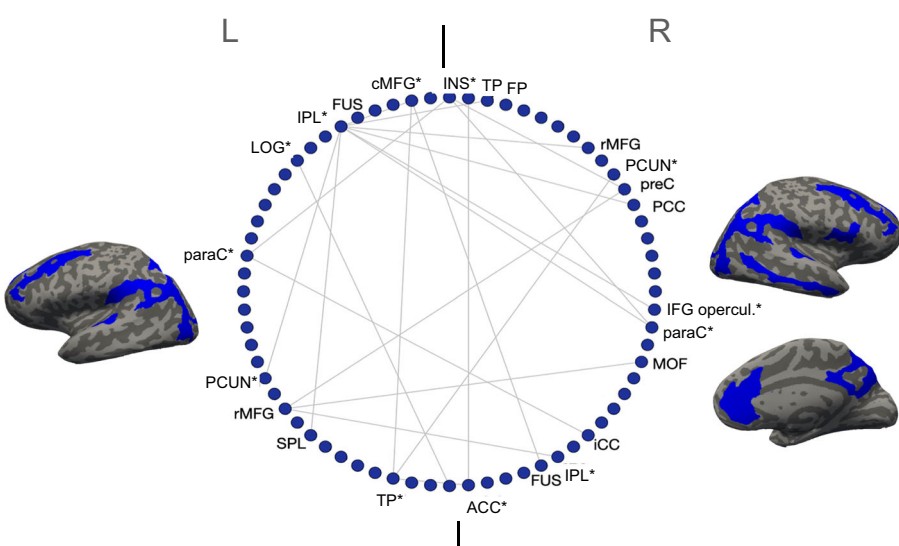

**Fig. 7 Regions with the most significant difference in inter-regional correlations of surface area between the groups: these correlations were significantly greater for PD-VH.** Only the inter-regional correlations with a difference greater than 0.3 in the $r^2$ are shown; the regions with the greatest number of significant inter-regional correlations, significant after multiple comparisons correction, are marked with a * (for details see S8). IPL inferior parietal lobule, SPL superior parietal lobule, LOG lat. occipital gyrus, MTG middle temporal gyrus, STG superior temporal gyrus, cMFG caudal middle frontal gyrus, PHG parahippocampal gyrus, paraC paracentral gyrus, preC precentral gyrus, IFG inferior frontal gyrus, SFG superior frontal gyrus, SMAR supramarginal gyrus, FP frontoparietal thickness, TP temporoparietal, FUS fusiform gyrus, CUN cuneus, PCUN precuneus, MOF middle orbitofrontal gyrus. The two vertical lines separate L and R hemisphere regions (left on left).

main sample and hippocampal reduction was found in the NPI subsample, after covarying for age, gender, TIV, disease onset, PD severity, medication, and cognition. This extends the postmortem literature, which has also identified the locations of Lewy body pathology in the basolateral nucleus of the amygdala associated with VH in PD patients at a similar level of cognitive impairment to those studied here[42]. Unlike the amygdala, there are only sparse Lewy bodies in the hippocampus at this disease stage and volume changes in this structure are more difficult to interpret. Other postmortem studies have indeed found more widespread cortical pathology at later stages of the disease[43]. Regions such as the occipital cortex have been found to have limited Lewy bodies even in late stages, and this is interesting considering that we found significant atrophy in occipital regions (for a broader discussion[44,45]). For this reason, we propose that it is more likely that visual hallucinations in these patients depend upon wider functional changes in brain networks and that this is related to neurotransmitters (e.g., loss of serotonergic or cholinergic projections and resulting cortical synaptic loss), rather than reflecting localised Lewy body neuropathology.

Since the prevalence of VH increases as PD progresses, tracking cognitive progression from PD-MCI to PD-dementia, it is difficult to disentangle brain changes related primarily to cognitive decline from those related primarily to VH or that may contribute equally to both. Reductions of hippocampal volume (particularly its anterior portions) have been found in some, but not all, studies of VH in PD, depending on whether patients are matched for cognitive decline[11,46]. Here, we found smaller hippocampi in the NPI sample where we were able to covary for age, gender, TIV, onset, LED, PD severity, and cognition. We did not find hippocampal volume reduction in the full data set covarying only for age, gender, and TIV. The volume reductions in the NPI analysis cannot be explained by differences in cognition or PD progression between groups, suggesting a role for the hippocampus in the mechanism of VH that is independent of

cognition[11], and highlighting the need to carefully design studies and control for cognitive and disease factors when examining hippocampal contributions to VH. The thalamus has been suggested as a key hub linking several cortical networks associated with VH in PD[18]. We did not find altered thalamic volumes in PD-VH in the main analyses or subgroup NPI analysis which included a wider range of covariates (see S4). This does not rule out the involvement of the thalamus in the pathophysiology of VH in PD but does suggest any functional changes in this structure are not associated with volume loss. Finally, reduced volume in cerebellar lobules VIII, IX/VII and Crus 1 is associated with VH in PD[47]. Freesurfer does not segment specific cerebellar subfields but volume changes were found in cerebellar white matter that may relate to these cerebellar cortical changes[47].

An earlier study of PD patients with visual hallucinations and MMSE > 25 (similar to our cohort) found only sparse Lewy body pathology in the cortex of PD patients with VH at the disease stage included in our analysis[43], raising the question of what causes the extensive cortical changes found in this and previous studies. One possibility is that such cortical changes represent synaptic loss secondary to degeneration in neurotransmitter inputs to the cortex. Previous studies have found changes in cholinergic, serotonergic, dopaminergic and GABAergic systems in PD patients with VH[27,28,48]; however, the relationship between regions of cortex with volume loss and the cortical distribution of these neurotransmitter systems has yet to be examined. We were able to investigate this relationship for subtypes of dopamine and serotonin receptors for which high resolution maps are available and found that cortical regions with higher binding had increased cortical volume loss, also taking into account spatial autocorrelation. The association, for 5-HT$_{2A}$ and 5-HT$_{1A}$ was confined to regions linked to VH rather than the cortex as a whole, suggesting the neurotransmitter effects were specific to VH, consistent with the possibility that degeneration in these neurotransmitter systems in PD underlies synaptic loss and cortical

## a) Surface area: Hubs

| Group | Hub Region |
|---|---|
| | |
| PD noVH | L lateral orbitofrontal |
| | L Superior frontal gy.* |
| | R MTG gy. |
| | L Fusiform |
| | L Lat. Occ. Gy. |
| | |
| PD VH | L Superior frontal gy.* |
| | R Superior frontal gy. |
| | R Middle Temporal gy. |
| | R Cuneus |

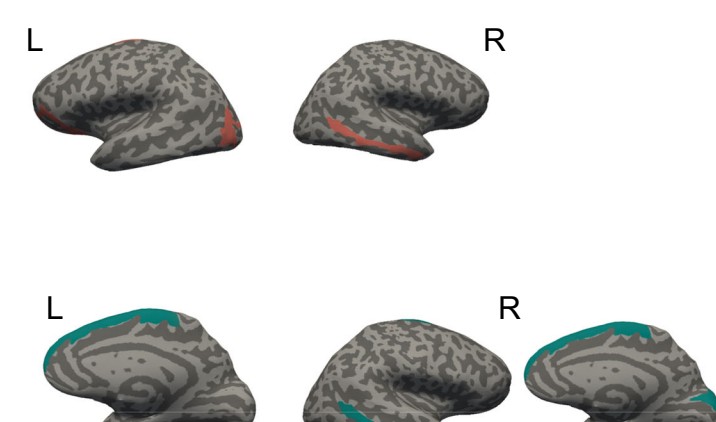

## b) Communities

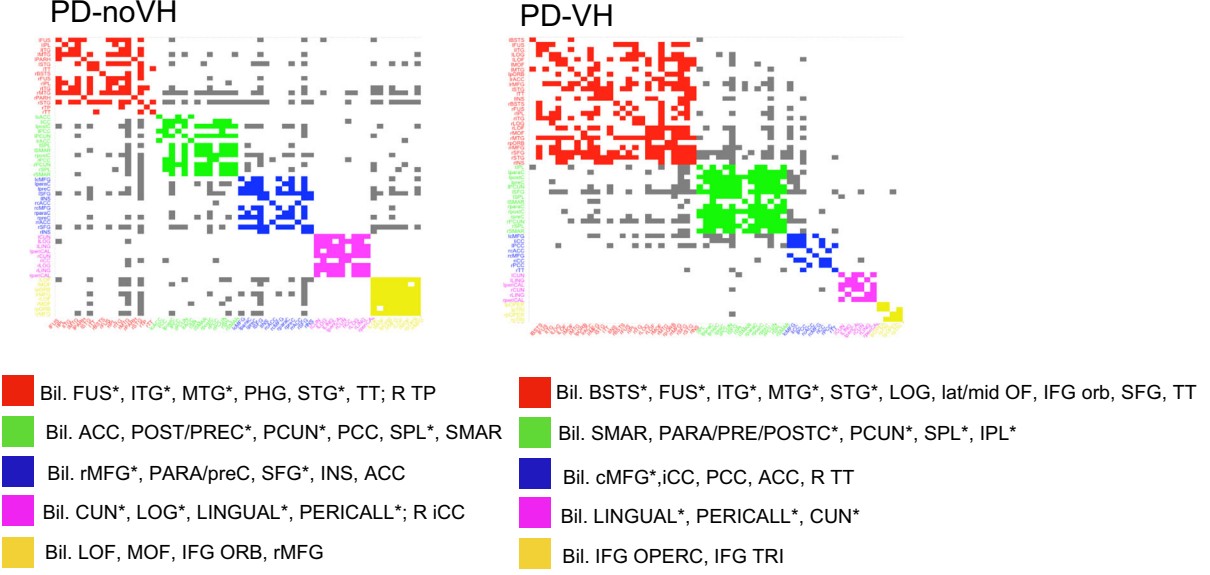

PD-noVH

■ Bil. FUS*, ITG*, MTG*, PHG, STG*, TT; R TP
■ Bil. ACC, POST/PREC*, PCUN*, PCC, SPL*, SMAR
■ Bil. rMFG*, PARA/preC, SFG*, INS, ACC
■ Bil. CUN*, LOG*, LINGUAL*, PERICALL*; R iCC
■ Bil. LOF, MOF, IFG ORB, rMFG

PD-VH

■ Bil. BSTS*, FUS*, ITG*, MTG*, STG*, LOG, lat/mid OF, IFG orb, SFG, TT
■ Bil. SMAR, PARA/PRE/POSTC*, PCUN*, SPL*, IPL*
■ Bil. cMFG*,iCC, PCC, ACC, R TT
■ Bil. LINGUAL*, PERICALL*, CUN*
■ Bil. IFG OPERC, IFG TRI

**Fig. 8 Hubs and communities: surface area. a** SA hubs identified based on efficiency, betweenness centrality and degree. Regions marked with a * are common hubs for both VH and noVH patients. **b** Communities identified for each group. Legend: red = 1st community, green = 2nd, blue = 3rd, pink = 4th, yellow = 5th. The regions identified for that same community also in the cortical thickness analysis are marked with a *. BSTS banks superior temporal sulcus, IPL inferior parietal lobule, SPL superior parietal lobule, LOG lat. occipital gyrus, MTG middle temporal gyrus, STG superior temporal gyrus, ITG inferior temporal gyrus, cMFG caudal middle frontal gyrus, PHG parahippocampal gyrus, paraC paracentral gyrus, preC precentral gyrus, postC postcentral gyrus, IFG inferior frontal gyrus, SFG superior frontal gyrus; SMAR supramarginal gyrus, FP frontoparietal thickness, TT temporal transverse, FUS fusiform gyrus, CUN cuneus.

thinning. Receptor binding maps for D2/D3 and 5-HT$_{2A}$ were not correlated suggesting different cortical regions contributed to the associations found for D2/D3 and 5-HT$_{2A}$. 5-HT$_{2A}$ and 5-HT$_{1A}$ binding maps were correlated so the same cortical regions are likely to have contributed to both serotonin findings. This finding was not specific to VH regions for dopamine, as we found it was a significant predictor when considering all regions as well as regions where the difference was not significant; so this may reflect a different process to the thickness alterations found specifically for VH. Finally, there was no direct suggestion of a greater contribution of one neurotransmitter system over the

other to cortical thickness loss, with equivalent slopes for all three receptors maps examined.

The examination of inter-regional correlations, with areas sharing reductions in thickness or surface area considered part of a functionally connected network, showed that regions of greater inter-regional thickness correlation in PD-VH overlap with those of the dorsal and ventral attention networks (DAN and VAN)[49], with the notable addition of para-hippocampal regions. Most of these regions of higher covariance have reduced thickness in PD-VH, suggesting the covariance is driven by correlated reductions in thickness. Dysregulation of VAN, DAN and default mode

networks (DMN) have been implicated in models of VH in PD[41] with reduced activity in the DAN of PD-VH[50], and the inter-regional covariance findings support this view. In contrast, the inter-regional SA covariance findings highlight key DMN regions in medial frontal and posterior cingulate cortex. These regions were not found to have reduced surface area in PD-VH, suggesting a relative preservation of the DMN compared to VAN and DAN. Indeed, results from dynamic fMRI have indicated active coupling between the DMN and the visual network, which correlated with the frequency of misperceptions, as opposed to reduced connectivity between the DMN, VAN and DAN[51]. Other metrics derived from the covariance structure include hubs defined by the richness of their interconnections and communities defined by their local strength of covariance. Hub metrics for thickness in the occipital lobe and parietal lobe were stronger in patients with VH, suggesting cortical thinning has a wider impact on the network in these patients, highlighting the importance of functional alterations in early visual areas in VH. One could argue that VH may not only depend upon on areas presenting neural pathology, but also on areas that may be relatively unaffected but operate in a network where there is pathology elsewhere, thus becoming functionally pathological while structurally intact[52]. Indeed, all the regions where richness of connections was either lower or higher for PD-VH fell outside areas of reduced SA in VH, suggestive of a more functional pathology which needs to be further explored with functional connectivity. Finally, there were qualitative differences in the communities of highly associated regions for PD-VH compared to PD-noVH in both the thickness and surface area analysis. Of particular note was the extent of interconnected areas in the ventral, lateral and medial temporal lobe that was larger in the PD-VH group. These regions had reduced thickness in PD-VH implying the local extent of thickness reduction is greater in PD-VH.

This mega-analysis of VH in PD pools data to create the largest sample of PD patients with and without VH tested to date. While this is a strength of the study, it also introduces complexities that smaller studies do not have to address. One is the variability of clinical data available for each site, limiting the analyses we could perform with the full dataset of 493 participants. This means that some of the key analyses, for example those related to cognitive covariates and disease duration or symptom scores, could only be carried out in smaller samples of 440 and 146 participants, but this is still substantially larger than any previous study. Another complexity is the need to address variance in the data caused by scanning at different sites and scanner types. Previous studies have typically used voxel-based methods to examine structural differences between PD-VH and PD-noVH. We used a different method to allow us to harmonise data between sites and examine cortical thickness and SA separately, but this means our findings are not directly comparable to those of previous studies. The primary focus of the study is on the cerebral cortex so we have not attempted to examine the detailed anatomy of regions such as the basal ganglia, hippocampus, cerebellum, and thalamus that may have a role in VH. We also do not have access to high resolution density maps for cholinergic receptor subtypes which limits the range of neurotransmitter analyses we can perform. Finally, as we do not directly measure VH in the scanner, we can only show links with the trait of VH, and not with the state of VH.

The mega-analysis has allowed us to resolve several uncertainties in the previous literature and describe relevant features of the VH phenotype in PD. With a sufficiently large sample, more widely distributed cortical involvement emerges than previously suspected with the finding of involvement of the primary visual cortex and its surrounds. Structural covariance modelling has

helped dissect out networks linked to attentional control within the widespread cortical regions affected, adding further evidence for the role of these networks in PD-VH. The findings also help resolve ambiguities between structural correlates of general cognitive decline or PD progression and those specifically related to VH. Patients at the same stage of PD and general cognitive impairment who experience VH have lower hippocampal volumes than those who do not. Our results may suggest a role for the hippocampus in models of VH in PD, although detailed analysis of hippocampal subdivisions is required before this can be substantiated. We are currently exploring this issue in a separate analysis. We can argue that the hippocampus represents part of an extended DMN composed of functional hubs, a dorsal medial subsystem and a medial temporal subsystem, which includes the hippocampus[53,54]. Thus, structural covariance, graph-level analyses and structural hippocampal imaging point to the involvement of attentional control networks in PD-VH. Finally, the findings shed light on why widespread cortical changes may occur at a stage of PD with only sparse cortical Lewy bodies. The associations between dopaminergic and serotonergic receptor binding and cortical thickness suggests that the cortical changes may be driven by neurotransmitter reductions with resulting cortical synaptic loss, raising the possibility of novel interventions to mitigate these effects at an earlier stage of disease. This is nevertheless a prediction from this data, longitudinal studies will be required to demonstrate this in the future.

## Methods

The study obtained King's College London ethical approval from Research Ethics Office, Psychiatry, Nursing and Midwifery (PNM) Research Ethics Panel (LRS-19/20-17680) on the 25/03/2020 and is pre-registered on the Open Science Framework site on 04/05/2020 (https://osf.io/nzatk). The methods follow the pre-registered plan with the addition of exploratory graph theoretical analyses based on structural covariance.

**Study selection**. We identified $N = 17$ studies/projects on VH in patients with PD that included acquisition of T1-weighted structural MRI scan, as part of a structural or functional data analyses, and with patients meeting our inclusion criteria (see below). We contacted the research groups responsible for the studies and among those $N = 8$ groups took part in the project, offering previously published and/or unpublished data: Prof. Simon Lewis (University of Sydney, Shine et al.[51] and unpublished data), Prof. Phil Hyu Lee and Dr. Chung (Yonsei University, Shin et al.[15]), Prof. Henry Mak, Prof. Grainne McAlonan and Prof. S.L. Ho (King's College London and The University of Hong Kong, Yao et al.[46]), Prof. Kathy Dujardin, Prof. Renaud Jardri and Dr. Delphine Pins (University of Lille, Lefebvre et al.[55]), Prof. John-Paul Taylor and Dr. Michael Firbank (Newcastle University, Firbank et al.[48]), Dr. Rimona Weil (University College London, sample in Zarkali et al.[56], T1-weighted data submitted), Prof. Michele Hu, Prof. Clare Mackay and Dr. Ludovica Griffanti (Oxford Parkinson's Centre Discovery Cohort, Baig et al.[57]; Griffanti et al.[58]), Prof. Dominic ffytche (King's College London, Lawn and ffytche[47]) (see Table 1 in the "Results" section for details). Only data from participants diagnosed as dementia-free were included to minimise the contribution to the study of global cortical changes in patients with PD dementia.

**Participants**. Raw T1-weighted MRI scans were obtained from eight different groups for a total of 519 subjects. We used 493 MRI scans in the analysis after discarding $N = 20$ participants who did not meet the criteria in terms of diagnosis (e.g., healthy controls, with diagnosis of dementia) or whose scan did not segment well during pre-processing and subsequent troubleshooting steps or was not suitable for analysis (e.g., motion) ($N = 6$). Patients with a MMSE score below 24 (raw) were retained ($N = 8$) only when part of a published work in which the absence of dementia was specifically stated. The final sample comprised 493 participants, 135 with VH (62 F, age = 67.85, SD = 7.74), 358 (131 F, age = 65.66, SD = 8.71) without VH (further details in "Results" section and in Table 1 and S2). Hallucination data collection varied across groups, as several used a different scale to screen for VH. Each group had previously divided patients into PD-VH and PD-noVH and we retained these original groupings for the mega-analysis.

**MRI data pre-processing and harmonisation**. MRI data was pre-processed with Freesurfer 6.0.0[59,60] to estimate cortical thickness, surface area and subcortical volumes. Data was processed on King's College London HPC infrastructure Rosalind (https://rosalind.kcl.ac.uk), with the standard recon-all procedure, consisting of motion correction, skull-stripping, affine registration to Talairach atlas,

segmentation, smoothing, and parcellation mapping. In order to screen for possible errors in the segmentation process, mean cortical thickness measures and manual slice by slice inspection were used to identify possible errors in the white-grey matter boundary and pial reconstruction steps. For subjects that did not segment properly the failed processing steps were re-run (autorecon3) after performing the appropriate corrections. Low quality scans (e.g., with excessive motion, $N = 4$) or scans that did not segment well upon troubleshooting ($N = 2$) were discarded. Individual cortical thickness, subcortical volumes and surface area measures were extracted based on the Destrieux atlas[61]. In order to explore structural differences between patients with and without VH across the different cohorts minimising variance due to different recruitment sites and, therefore, different scanners, we used a harmonisation method. ComBat is an empirical Bayesian algorithm aiming at minimising the variance due to the scanner features and to maintain the variance related to biological features and has been previously successfully used in studies of cortical thickness[62,63]. In this study, this method has been also used to harmonise volume and surface area for each participant (see Supplementary Note S1 for more details about this method and plotted results).

**Group differences analysis**. First, we conducted a meta-analysis with R package "metafor"[64] to check whether patients differed the relevant demographic and clinical variables. Results are mentioned in the main text and reported with forest plots and a detailed description in Supplementary Note S2a.

Then, we conducted separate exploratory ANOVAs to reduce the number of regions to be entered in the MANCOVAs with age, gender, and total intracranial volume (TIV) as covariates. The ANOVAs results were corrected for multiple comparisons (for number of regions entered in the MANCOVA $N = 148$ for thickness, $N = 148$ for surface area, $N = 21$ for subcortical volumes) were corrected with the Benjamini-Hochberg procedure ($p_{FDR}$). Only regions surviving this correction were entered in the MANCOVAs for cortical thickness, surface area and subcortical volumes to screen for group differences between patients with and without hallucinations, with age, gender and TIV as covariates. Pairwise comparisons between the two groups were Bonferroni corrected (see tables in Supplementary Note 3). The models were calculated using SPSS 24.0.0.0 (IBM corp. 2016) and R 4.0.0 (R core team, 2017). Results are presented in Fig. 1, created with a custom colour coding based on effect size and by overlaying region labels on a brain render.

We used Tukey's method programmed in R with the 1.5*IQR rule to identify outliers other than those excluded upon unsuccessful pre-processing. This allowed the careful inspection of the identified subjects in order to verify whether the outlier value depended upon measure errors (e.g., harmonisation bugs) or incorrectly entered data, or on the subject, with the purpose of retaining outliers depending on the subject (e.g., intrinsic features of the subject). No participants were discarded upon this check for this analysis.

**Sensitivity and subgroup analysis**. Of the eight original groups, three used the Neuropsychiatric Inventory (NPI) to score visual hallucinations. For this subgroup of studies, patients did not differ on age, gender, onset, levodopa equivalent daily dose (LEDD), and Mini Mental State Examination (MMSE) score. Within each of the three original studies, patients were matched in terms of motor symptoms severity (UPDRS-III). We ran a one-way ANOVA to check for differences on the UPDRS-III, but data was missing for 20 participants. We computed the group mean and used that to fill the missing values for the between groups multivariate ANOVA. We carried out partial correlations controlling for LED, UPDRS-III, MMSE and age between NPI score and the cortical thickness, surface area and subcortical volume data. In addition, we compared the PD-VH and PD-noVH (35 females PD-noVH, 27 females PD-VH, age PD-VH = 70.39 ± 6.82, PD-noVH = 69.64 ± 6.45) in the data set using the original VH binary scores to check for consistency in the results with the larger data set, including age, gender, disease onset, LED, PD severity (UPDRS-III) and MMSE as covariates (Supplementary Note 4 and Fig. 1b). We also conducted analyses of variance with a larger subgroup and with graded VH scores (mild, moderate, and severe), together with an ordinal logistic regression (for details on the sample, methods and results see Supplementary Note 5).

We ran the same MANCOVAs for thickness, surface area and subcortical volumes in a subsample of 440 individuals (319 noVH, with 116F, age = 65.9 sd = 8.86; 121 VH with 57F, age = 68, sd = 7.89) for which additional data (UPDRS-III, MMSE, LED, disease onset) was available. All demographics and clinical details are reported in S3b. The purpose was to assess the robustness of the results taking into account the variability associated not only to age, gender, and TIV. Results are consistent with those reported for the main sample and are reported in Supplementary Note 3b.

In addition, with the scope of further assessing the robustness of the results, we performed the same MANCOVA used for the main sample, using a leave one (group) out approach: the model was carried out for all groups minus one (seven models), allowing to analyse the consistency of the group differences when removing each of the groups. Results are reported in Supplementary Note 3c, after the tables describing the results of the main analyses.

**Receptor density profiles**. Regression models with the difference of the means (VH−noVH) of morphometrical features (thickness, subcortical volume) as

dependent variable and receptors density profiles as predictors were carried out, with a methodology similar to Selvaggi et al.[65]. Specifically, receptors density profiles were obtained for D2/D3, 5-HT$_{1A}$ and 5-HT$_{2A}$ based on a [18F] Fallypride template[66] and a [11C] Cumi-101 5-HT$_{1A}$ and a [11C] Cimbi-36 5-HT$_{2A}$ templates[67], respectively. These templates have been developed on PET data from healthy participants and thus constitute a measure of pre-morbid receptor density distribution. We have focussed on DA and 5-HT as high resolution templates are available for these receptors of interest at the moment. Including cholinergic maps in the analysis would greatly enrich this approach given the importance of cholinergic transmission in VH in PD[68], and will be done once high resolutions templates will be available. [18F] Fallypride is a D2/D3 receptor antagonist with a high signal to noise ratio[69]. [11C] Cumi-101 and [11C] Cimbi-36 are high affinity PET radioligands for 5-HT1$_A$ and 5-HT$_{2A}$ receptors[67]. Parametric modelling of the binding potential used the cerebellum as reference region[70] and thus the vertices corresponding to the cerebellum were excluded from the regression analyses. Each of these templates was registered to the Talairach space using the *fsaverage* template subject and parcellated with the Destrieux atlas, to ensure alignment with the parcellated structural data of our participants. For each of the vertices we extracted the binding potential using *fslmeants*. Regression models were carried out to estimate the relationship between cortical thickness and surface area differences of the mean between VH and noVH patients (regions resulting from the first group-level MANCOVAs and ANOVAs, see Results and Supplementary Note 3) and receptor density profiles. For each receptor we ran three different models. First, we examined the relationship between the receptor's binding potential in the regions with significant differences in cortical thickness area between PD-VH and PD-noVH. The slopes for these models were also compared (Supplementary Note 6). Then, in order to better investigate such relationship, we also assessed whether the receptor's binding potential could predict thickness values for all regions; finally, with the same purpose, we ran models considering only regions where the difference between the groups was not significant (Supplementary Fig. 3, Supplementary Note 6). With the scope of taking into account the role of spatial autocorrelation—the fact that neighbouring data points in the brain are not statistically independent—we also ran correlational analyses following the method described in Vâša et al.[71]. The choice to use the Vâša method was primarily determined by the fact that this method applies spatial permutations to parcellated data. In particular, $N = 10,000$ permutations of the regional coordinates were generated. These permutations were then used in a correlation analysis between each of the PET based maps and the regional difference of the means, the same values used in the regression models carried out. Since this model requires a symmetrical number of regions per hemisphere to produce the permutations, we carried out this analysis for cortical thickness in all regions, and for a subset of common regions for the analysis with regions shown to differ with the MANCOVA. For a broader discussion on spatial autocorrelation in brain imaging analysis and a comparison of the different methods, see ref. [72].

We also ran an exploratory model for subcortical volumes including all regions only, as the results were restricted to the bilateral amygdala (see Supplementary Note 6). Linear regression models were coded in R using the packages rstatix 0.7.0[73] and MASS 7.3-54[74]. For each regression model, in order to identify outliers, Cook's distance was computed and any data point with a Cook's distance >1 was marked as highly influential, explored and if appropriate discarded[75]. In addition, the confidence intervals of the significant regression models were estimated with the bootstrapping technique[76] with 10,000 cycles. Methods and results for thickness and surface area are graphically represented in Figs. 2 and 3, for results on volume and further details see Supplementary Note S6. Scatter plots were created using ggplot2 3.3.5.

**Principal component analysis (PCA)**. Results from the MANCOVAs comparing PD-VH and PD-noVH highlighted the involvement of widespread cortical regions in a high dimensional dataset. We used PCA, in order to reduce the dimensionality of the dataset and to identify putative latent dimensions underlying the differences in structure in PD-VH versus PD-noVH patients while retaining as much variance as possible (Joliffe and Cadima[77]). We entered data from both hemispheres. Analyses were carried out with R packages *factominer* 2.47[78] and *factoextra* 1.0.7[79]. The scree plot for the PCA is reported in Supplementary Note 7 and Supplementary Fig. S4. PCA inputs comprised the significantly different regions from the MANCOVA (see Supplementary Note 3 for a list). Results are presented in Fig. 4, created with a custom colour coding based on the components and by overlaying region labels on a brain render.

**Structural covariance and graph theory analysis**. To investigate inter-regional properties to explore and characterise the grey matter network-level organisation of PD-VH, we built networks based on structural covariance, a technique that assays covariation of differences in grey matter morphology between different brain structures across a specific population[80,81]. Since the most widely used atlas for this kind of analysis is the Desikan-Killiany atlas[82,83], we extracted morphometric features (thickness, surface area) at the 68 vertices of this atlas. The dataset was harmonised for multi-site effects with the same procedure described in section 4.3.1. The dataset was reduced to 467 cases as the design matrix based on the full dataset was not invertible due to high collinearity of some columns. We discarded

$N = 26$ subjects coming from the smallest datasets and the problem was over-ridden. Homogeneity of groups was verified with a Levene's test[84,85].

The dataset counts 467 subjects, 118 PD-VH (56F), mean age = 67.19, sd = 7.77, 349 PD-noVH (129F) mean age = 65.7, sd = 8.77, with participants not differing on age. *Age* and *gender* were used as covariates in the models. Analyses were carried out with R package *braingraph* version 2.7.0[86,87] and *igraph* 1.2.7 (Csardi and Nepuz[88]). To construct the networks, first we specified a general linear model for each region (thickness/area as outcome variable, age, and gender as covariates). The structural covariance matrices of the two groups were defined by estimating the inter-regional correlation between model residuals of thickness and area (in separate models)[89] to build a $68 \times 68$ matrix. Inter-regional correlation coefficients were compared with the *cocor* method in R, and *p* values for such comparisons were subsequently corrected with the Benjamini–Hochberg method. A density-based threshold[90] was applied to the matrix in order to retain a percentage of the most positive correlations as non-zero elements in a binary adjacency matrix. Different densities ranging from 0.05 to 0.20 with a 0.01 step size were explored. The differences between PD-VH and PD-noVH covariance matrices were then computed, first to establish that the two matrices differed significantly from one another; secondly, a cell-by-cell comparison was carried out to establish which covariance patterns were significantly greater for the PD-VH group compared to the PD-noVH group. Random undirected and unweighted graphs were created for each group, and vertex-level and graph-level metrics were computed for each group. For visualisation purposes a density of 0.13 was selected. Vertex importance was assessed using degree, betweenness centrality and nodal efficiency. A hub was categorised as such if its betweenness centrality was greater than the mean plus 1 standard deviation—calculated on all vertices at the same density (e.g., [86,91–94]) - for at least half of the densities. To assess network segregation in order to better understand the communities observed, we used modularity, which is a measure of the strength of network partitions. High modularity is a measure of how much vertices from the same community are more connected to each other. Modularity was computed with the Louvain algorithms, which also partitioned the network in communities[95]. Cortical thickness-based networks have been shown to have distinct modules/communities of regions, similar to those derived from fMRI and DTI data (see ref. [86]). Network analyses were performed with permutation tests (5000 cycles) and bootstrapping analyses to compare vertex-level measures. (Figs. 5–8 and Supplementary Note 8).

Finally, to further assess the relationship between graph level metrics and visual hallucinations in the full sample, we computed Pearson's correlation coefficients between the difference of the means of graph metrics of interest (local and nodal efficiency) for the models on thickness and surface area separately, and the difference of the means in thickness and in surface area, respectively, with the NPI subsample, for which we have all clinical and demographic information and in which participants do not differ on all those variables. All results were false discovery rate corrected.

**Reporting summary**. Further information on research design is available in the Nature Research Reporting Summary linked to this article.

## Data availability

Source Data are provided as Source Data file, together with the [$^{18}$F] Fallypride template. The template and source data are also available on the project page on the Open Science Framework website: https://osf.io/fv2k7/files/. 5-HT maps: https://xtra.nru.dk/FS5ht-atlas/. Further information and request for resources should be directed to and will be fulfilled by the lead author Miriam Vignando (miriam.vignando@kcl.ac.uk). This study did not generate new unique reagents. Source data are provided with this paper.

## Code availability

We did not generate new ad hoc code for the study, as all the analyses were based on pre-existing R packages or publicly available codes. We provide the R codes generated for the structural covariance analysis (R package braingraph 2.7.0), for the principal component analysis (R package factominer 2.4) and the receptor density maps models (R package rstatix 0.7.0 (and MASS 7.3-54).

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

## Acknowledgements

Dr. Miriam Vignando was supported by the MRC grant MR/R005931/1 awarded to Prof. Mehta. This paper represents independent research part supported by the National Institute for Health Research (NIHR) Maudsley Biomedical Research Centre at South London and Maudsley NHS Foundation Trust and King's College London. The views expressed are those of the author(s) and not necessarily those of the NHS, the NIHR or the Department of Health and Social Care. Prof. ffytche was supported by NIHR Programme Grants for Applied Research (RP-PG-0610-10100) and the South London and Maudsley NHS Foundation Trust Mental Health BRC. Prof. John-Paul Taylor and Dr. Firbank's study was supported by the Northumberland Tyne and Wear NHS Foundation Trust, and by the National Institute for Health Research (NIHR) Newcastle Biomedical Research Centre (BRC) in Newcastle upon Tyne Hospitals NHS Foundation Trust and Newcastle University. Prof. Simon Lewis wishes to thank the NHMRC for the Dementia Research Team Grant, NHMRC Project Grant, Parkinson's NSW Seed Grant 2013, NHMRC Investigator grant, NHMRC-ARC Dementia Fellowship, NHMRC Practitioner Fellowship. Prof. Hu, Prof. Mackay and Dr. Griffanti wish to thank the Monument Trust Discovery Award from Parkinson's UK, the MRC Dementias Platform UK and the National Institute for Health Research (NIHR) Oxford Biomedical Research Centre (BRC), and the NIHR Oxford Health BRC. Prof. Seok Jong Chung wishes to thank the National Research Foundation of Korea funded by the Ministry of Education (Grant NRF-2018R1D1A1B07048959). Dr. Rimona Weil wishes to thank the Wellcome Trust and the UCLH Biomedical Research Centre. Prof. Henry Mak wishes to thank the State Key Laboratory of Brain and Cognitive Sciences, The University of Hong Kong.

## Author contributions

Conceptualisation, M.V., D.F., and M.M.; Methodology, M.V. and M.M.; Investigation, M.M., M.V., D.F.; Formal analysis, M.V.; Writing—Original draft, M.V., M.M., and D.F.; Writing—Review and Editing, P.H.L., S.J.C., S.L., R.W., L.G., M.H., C.M., G.M., H.M., D.P., R.J., K.D., J.P.T., M.F., and S.L.H.; Visualisation, M.V.; Data curation, M.V.; Funding acquisition, M.V. was supported by MRC grant MR/R005931/1 awarded to M.M. and D.F.; Resources, P.H.L., S.J.C., S.L., R.W., L.G., M.H., C.M., G.M., H.M., S.L.H., D.P., R.J., K.D., J.P.T., M.F., and D.F.; Supervision, M.M.

## Competing interests

Dr. Rimona Weil has received speaker honoraria from GE Healthcare. Prof. Hu reports funding/grant support from Parkinson's UK, Oxford NIHR BRC, University of Oxford, CPT, Lab10X, NIHR, Michael J Fox Foundation, H2020 European Union, GE Healthcare and the PSP Association. She also received payment for Advisory Board attendance/ consultancy for Biogen, Roche, CuraSen Therapeutics, Evidera, Manus Neurodynamica and the MJFF Digital Health Assessment Board, outside the submitted work. J-P.T. has received speaker honoraria from GE Healthcare and acted as consultant for Kwoya-Kirin and received funding from Sosei-Heptares. These have no association with the present publication. J-P.T. reports funding/grant support from Newcastle NHIR BRC based at Newcastle upon Tyne Hospitals NHS Foundation Trust and Newcastle University. The remaining authors declare no competing interests.
