## [Peer Review File · Nature Communications]

Mapping brain structural differences and neuroreceptor correlates in Parkinson's disease visual hallucinations: a mega-analysisREVIEWER COMMENTS

Reviewer #1 (Remarks to the Author):

Review

This study is investigating the pathophysiology of Visual Hallucination (VH) in PD by using brain imaging changes that may correlate with (VH) in subjects with PD. To date most studies have been single centre with small numbers of subject that preclude conclusions. The authors have performed a mega analysis in 493 PD subjects using clinical and imaging data, and have correlated regional cortical thickness, surface area changes to clinical features between subjects with VH (n = 135) and without (n = 358). Subjects were matched for age, gender, disease onset, MMSE, UPDRS-III and levodopa equivalent daily dose (LED). The study shows widespread decreased cortical thickness in most cortical areas in PD-VH; greatest effect size were medial occipital parietal and frontal regions. There was also a right sided predominance. Surface area was reduced in PD-VH in the left and right medial occipital and in the left insular gyrus, and in the medial central and superior frontal regions. Also showed that correlating the NPI hallucinations subscore (a measure of psychosis although not specific for PD) with morphometrics, there was significant inverse correlations for right hemisphere cortical thickness in the intraparietal sulcus ; the superior temporal sulcus, the Jensen sulcus (between the anterior and posterior rami of the IPS) and the cingulum (marginalis) and significant positive correlation was found with the right frontomarginal gyrus. Neuropharmacology of DA and 5HT with VH was also assessed via structural differences are related to the spatial variation in subtypes of receptors for which high resolution PET atlases are available (dopamine and serotonin). 5HT2A receptor loss did appear to correlate with VH which fits with some of the current pharmacological agents used to treat VH in PD. The study is important and adds to knowledge about possible pathophysiology of VH.

The conclusions appear robust and the discussion covers shortcomings of the data. However, also worth emphasising one challenge with studying VH in PD, is that the patient is usually not actively hallucinating in the scanner – and rather what is measured is possibly / inferred to be linked to VH.

Proof read eg line 1 'Patients with Parkinson's disease (PD), aside the typical motor symptoms' Line 6: '....progress to formed hallucinations (initially with insight preserved), then hallucinations in other modalities and delusions'. I disagree that the modality of the hallucination changes – as the authors seem to have implied. I have never known a PD patient to switch from one sense to another

Minor point – maybe shorten the manuscript

Reviewer #2 (Remarks to the Author):

This study deals with the structural and neuroreceptor correlates of Parkinson's disease psychosis, mainly visual hallucinations. Authors claim that previous imaging studies investigating the neural correlates of visual hallucinations (VH) in PD have been relatively heterogeneous in their findings due to differences in study design and limitations of scale. Then, to overcome these limitations, the authors use an interesting statistical approach (empirical Bayes harmonisation) to pool together structural imaging data from multiple research groups into a large-scale mega-analysis to identify cortical regions and networks involved in VH and their relation to receptor binding. Using this methodological approach to study cortical thickness the authors claim to have found a wider cortical involvement underlying VH than previously recognised, including primary visual cortex and its surrounds, and the hippocampus, this latter being a structure that does not currently play a central role in models of VH. By interpreting the results of the structural covariance analyses they suggest a strong involvement of the attention control networks in VH. Associations found between serotonergic and dopaminergic receptor binding and cortical thickness move the authors to claim that the present study provides the first evidence that the cortical changes may be driven by neurotransmitters reductions. If confirmed, this latter finding rise the possibility of novel

interventions to compensate for the neurotransmitter loss in patients suffering VH at an earlier stage of disease.

The study is interesting and the conclusions are original and potentially of interest and influence to others in the scientific and practicing community.

My main doubts and concerns are methodological and can probably be addressed by the authors:

1. I could not find any reference on how did the authors handle partial volume correction when working with PET data. That is, PET uptake should always be regionally corrected for partial volume effects, i.e for possibly-reduced gray matter density. In other words, finding a direct correlation between cholinergic/serotonergic PET uptake in a region with its cortical thickness or volume in the same region might simply be a consequence of reduced neural density in that region (and therefore lower PET uptake). This would not imply that atrophy in that region is caused by a cholinergic/serotonergic deficit, as neural loss could have been caused by another pathological process and these correlations would still hold due to partial volume effects. Therefore, caution must be taken when concluding about this type of associations.

2. The authors state that "Results were false discovery rate corrected", but do not specify the corrected p-threshold. Some results are reported at $p_{\text{fdr}} < 0.09$, which may not reach significance.

3. How did the authors corrected for the fact that multiple graph metrics were considered (vulnerability, transitivity, local and nodal efficiency, path length, betweenness centrality, eccentricity, distance) and correlated with multiple structural metrics (surface area, thickness...) in multiple samples? In this setting, mild correlations reported as significant such as the one between surface area and local efficiency ($r=0.24$, $p=0.02$) could be questioned.

4. The authors state that "The hippocampus does not currently play a central role in models of VH in PD and our findings suggest this needs to be reconsidered". As they have already ran FreeSurfer into the T1-MRI scans, the authors could have also included in the analysis the anatomical parcellation of the hippocampus that FreeSurfer provides (<https://surfer.nmr.mgh.harvard.edu/fswiki/HippocampalSubfieldsAndNucleiOfAmygdala>). This would have allowed improved anatomical resolution in this important region.

5. Other works in the field, albeit using VBM-GM instead of surface-based techniques, report cortical alterations (even in minor hallucinations) that are congruent with the overall results of this work. I suggest commenting them: doi: 10.1002/mds.27557, <https://doi.org/10.1111/ene.14576>, <https://doi.org/10.1101/2020.05.11.054619>

Reviewer #3 (Remarks to the Author):

The authors should be congratulated on the efforts of assembling the clinical and morphometric imaging data from 8 studies of Parkinson's patients with and without visual hallucination. Compared with a meta-analysis based on individual small studies and inconsistent findings, the final dataset of 493 subjects would allow enhanced statistical powers to identify group differences between PD-VH and PD-noVH. Statistical harmonization using ComBat was performed to remove site differences on the cortical thickness and surface data from 68 vertices of the Desikan-Killiany atlas. Statistical analyses were carried to compare group differences in mean and covariance of the morphometric features, and to correlate with clinical scores for visual hallucination and receptor binding. The paper is overall nicely written but might benefit from some more clarifications and details. In particular, I have the following comments:

1. Clinical characteristics: Table 1 provided p-values testing PD-VH vs. PD-noVH within each study. It would also be helpful to compare the distributions of the clinical characteristics across 8 studies using e.g. boxplots. In addition, since gender was adjusted as one of the covariate, I suggest the authors to report the proportion of female as a separate column and test for difference using e.g. Chi-square test.

2. Harmonization: since each study contain both VH and no-VH participants, was site differences

only estimated using the no-VH patients, or was VH diagnosis adjusted in the ComBat model. In addition, is there any need for harmonization on the clinical outcome such as NPI?

3. For group comparison using ANOVA, Bonferroni correction was used for multiple comparison. Was each corrected for 68 regions in each analysis? Bonferroni seems could be quite conservative given the number of regions to be corrected.

4. ANOVA and Figure 1, rather than only display p-values on the brain regions, it would be more informative to indicate the effect size. Similarly, it would be interesting to examine the direction of age association and effect sizes as well. This might be achieved by running a linear regression adjusting for age, gender and ITV.

5. Receptor density analysis: I'm confused as why the estimated mean difference of the morphometrical features was used as dependent variable. Were there only one set of receptor density value per group? If I understand correctly, each dot on Figure 2 b) corresponds to a region. If so, are the magnitude of the mean difference across cortical regions comparable? Do they need to be normalized to obtain meaningful interpretation of the correlation between mean difference cortical thickness and receptor density?

6. PCA analysis: unless sparse PCA is performed, it is not clear how representative those claimed 'best representative' regions are compared to rest of the regions and what are the directions of association. In fact, it will be helpful to plot the contributing weights of PC 1&2 from each of the regions, or alternatively conducting post-hoc analyses to correlate the identified regions with PC scores or NPI.

7. The conclusions of significance correlation between PC loadings and NPI is quite strong given that p-values are quite marginal ($p=0.046$ and 0.049) and not sure if multiple comparison correction was done here.

8. Network analysis: how was cell-by-cell comparison carried out since there is only one estimated structural covariance matrix for PD-VH and PD-noVH? and How robust was the conclusion under different the choice of graph density?

Minor:

1. Would suggest to use the term 'negative correlation' rather than 'Inverse correlation'

Reviewer #4 (Remarks to the Author):

The authors combined several existing and new datasets to attempt to map the pattern of brain atrophy associated with visual hallucinations in Parkinson's Disease (PD). All included studies had the same goal but suffered from small sample sizes, making the current mega-analysis pertinent.

A key question is whether the theory, as assumed in some of the literature, that psychosis and hallucinations in PD are simply a result of widespread cortical involvement plus the use of dopaminergic medications is correct. An alternate theory would be that hallucinations are the result of involvement of specific cortical areas and/or intrinsic networks. To this end they compared structural T1-weighted MRI scans of patients with and without visual hallucinations (PD-VH vs PD-noVH) using Freesurfer to measure cortical thickness, cortical surface area, and subcortical brain volumes. They used Bayesian harmonization to correct for site. The final sample was 135 PD-VH and 358 PD-noVH.

The main findings are as follows: widespread lower cortical thickness (CT) in the VH group. This seemed to encompass almost the entire cortex, with a few exceptions; lower volume of the amygdala in the VH group; some correlations between lower CT and a Neuropsychiatric Inventory of hallucinations in a subgroup. There is also an attempt to compare the cortical areas affected to maps of dopamine and 5HT receptor distributions in healthy subjects. Here it seems there was no relationship for 5HT when considering the entire cortex, but an inverse relationship if only cortical areas showing an VH-noVH effect are considered. This implies that higher 5HT innervated areas

showed greater relative atrophy in the VH group. Findings for dopamine receptors were not significant. Finally, additional analyses with PCA and structural covariance more or less confirmed the initial cortical thickness analysis.

The main strength of this work is the collating of a very large dataset from multiple sites, processing the data using the same tools in one center, trying to match for age and disease stage, and using rigorous techniques to account for site effects.

However, I have a few concerns regarding the authors' interpretations and a significant concern regarding the significance of the result. Specifically I do not see how they have achieved their stated goal of: "... [disentangling] brain changes related specifically to VH mechanisms as distinct from those related to cognitive decline, PD stage or medication effects".

The most significant concern is that it is not clear that the groups are adequately matched – although the authors did make every effort to check and account for any discrepancies. In a PD sample, two factors will overwhelmingly determine cortical thickness and subcortical volumes: age and disease stage. It is difficult to tell, as some of the information is in the supplementary materials, but it seems that the VH group is older and at a more advanced stage of the disease based on UPDRS and MMSE. I am trying to ascertain if there are differences based on Table 1 and Fig. S2, but these do not appear to always match up. Nonetheless it would seem that the VH group was older and had worse MMSE (indicative of worse disease). For UPDRS, it is not clear as the forest plot in S2 shows a negative total value, but contrasting it to Table 1 would indicate higher UPDRS scores in the VH group? In any case this part of the results needs to be clearer, and placed in the main text as it is crucial. Finally, it says in the supplementary materials that "we did not have raw data for all groups for all relevant variables,...", which makes it difficult to see whether or not the groups were truly matched. Obviously, a lack of a $p < 0.05$ difference does not mean a lack of important effect. Also it is not necessarily the case that putting age as a confound will eliminate the bias.

Many in the dementia field are now using the W-score approach to account for age and sex effects on MRI measures – as outlined in *Journal of Neuroscience* 32(46):16265-16273, but this would require healthy controls scanned at the same sites.

Questions regarding the Interpretations:

1. It is not clear what the main conclusion of the study is. If it is that VH is associated with greater cortical loss, this is not surprising. As mentioned by the authors, psychosis and hallucinations are a harbinger of dementia and loss of autonomy, likely indicating widespread pathology. There is an extensive postmortem literature on visual hallucinations in PD, going back decades, which consistently associates this symptom with greater Lewy pathology in limbic and cortical areas.
2. The PCA analysis does not seem to add anything. It seems a more complicated analytical approach that also concludes in reduced cortical thickness in the VH group. A similar argument can be made for the structural covariance analysis. In this analysis only two thresholded correlation matrices (one per group) are compared with a wide array of network measures. The conclusion that the group differences especially involve the dorsal and ventral attention networks is difficult to follow since it is not supported by the more straightforward original group differences analysis. A systematic analysis of cortical thickness differences for each intrinsic network might better prove this point.
3. The conclusion that degeneration in 5HT neurons may lead to cortical degeneration in projection sites is not at all supported by this study – as this was not the point of the experiment. An analysis with control subjects' MRI data at the minimum would be needed to assert that 5HT projection sites are more vulnerable in PD. But even then, this would not implicate loss of 5HT projections in cortical neurodegeneration, as 5HT innervation was not measured in this study. Finally, this disregards the evidence that PD is caused by synucleinopathy that affects many neuronal types.

Minor concerns:

1. The authors state that there is only "sparse Lewy pathology in the cortex of PD patients with VH at the stage included in our analysis". The citation to Harding et al. *Brain* 2002 is inappropriate as this is a study of amygdala pathology only. Moreover, there have been many postmortem studies showing diffuse cortical Lewy pathology in patients with hallucinations.

2. I could not tell if UPDRS-III was on or off medications.

Point by point response to reviewer's comments:

We thank all reviewers for their comments and the suggestions provided. Below we address each comment in turn.

In addition, we have taken the opportunity of the revision to further reflect on some of the Reviewer's comments and our analyses, making three refinements which confirm the robustness of the findings:

1. For the **group-level** analysis: initially for the MANCOVAs we used a 2-stage approach: a feature selection one-way ANOVA, and a MANCOVA using the regions screened with the ANOVA. Multiple comparisons correction was applied on the MANCOVA results. We have decided to use a more conservative approach: we have applied multiple comparisons correction (FDR) after the feature selection stage with the 148 regions, and only the regions surviving this correction were then entered in the MANCOVAs. This gives a slightly reduced set of regions for cortical thickness, thus we have repeated the PCA, the receptors regression models and the sensitivity analysis on the NPI subsample. The overall findings are largely unchanged with some small changes in the statistical values. However, the results for surface area are reduced to one region, and thus surface area has been removed from PCA and receptor analyses.

2. In light of another recent manuscript completed within our department, we now take into account **spatial autocorrelation** for the use of receptor density profiles. The Methods described this approach: "With the scope of taking into account the role of spatial autocorrelation – the fact that neighbouring data points in the brain are not statistically independent - we also ran correlational analyses following the method described in Váša et al., 2018. The choice to use the Váša method was primarily determined by the fact that this method applies spatial permutations to parcellated data. In particular, N=10000 permutations of the regional coordinates were generated. These permutations were then used in a correlation analysis between each of the PET based maps and the regional difference of the means, the same values used in the regression models carried out. Since this model requires a symmetrical number of regions per hemisphere to produce the permutations, we carried out this analysis for cortical thickness in all regions, and for a subset of common regions for the

analysis with regions shown to differ with the MANCOVA. For a broader discussion on spatial autocorrelation in brain imaging analysis and a comparison of the different methods, see Markello and Masic, 2021.”

3. In order to further assess the robustness of the results, we performed the same MANCOVA described in 4.3.1 using a leave one (group) out approach: the model was carried out for all groups minus one (7 models), allowing us to assess the consistency of the group differences when removing each of the groups. Importantly, the results are consistent across groups with minor differences, reported in **Supplemental Information S3c**.

4. Following some reviewers’ concerns we have also added a group-level MANCOVA with age, gender, TIV, onset, LED, MMSE, UPDRS-III as covariates for a subsample of 440 patients for which we had such information (there were some missing values – replaced with the mean for that variable for the specific study. N of missing values is reported in SI 3b together with results). The purpose of this analysis was to test the robustness of the model on the main sample taking into account relevant covariates for which each study had matched their participants but that were not matched in our full sample. Results are strongly consistent with those from the main model and are reported in the main text and with greater detail in S3b. We have also updated the correlational analysis between NPI and cortical thickness, now controlling for age, MMSE, LED and UPDRS-III (main text 2.3).

Reviewer #1 (Remarks to the Author)

Review

This study is investigating the pathophysiology of Visual Hallucination (VH) in PD by using brain imaging changes that may correlate with (VH) in subjects with PD. To date most studies have been single centre with small numbers of subject that preclude conclusions. The authors have performed a mega analysis in 493 PD subjects using clinical and imaging data, and have correlated regional cortical thickness, surface area changes to clinical features between subjects with VH (n = 135) and without (n = 358). Subjects were matched for age, gender, disease onset, MMSE, UPDRS-III and levodopa equivalent daily dose (LED). The study shows widespread decreased cortical thickness in most cortical areas in PD-VH;

greatest effect size were medial occipital parietal and frontal regions. There was also a right sided predominance. Surface area was reduced in PD-VH in the left and right medial occipital and in the left insular gyrus, and in the medial central and superior frontal regions.

Also

showed that correlating the NPI hallucinations subscore (a measure of psychosis although not specific for PD) with morphometrics, there was significant inverse correlations for right hemisphere cortical thickness in the intraparietal sulcus ; the superior temporal sulcus, the Jensen sulcus (between the anterior and posterior rami of the IPS) and the cingulum (marginalis) and significant positive correlation was found with the right frontomarginal gyrus. Neuropharmacology of DA and 5HT with VH was also assessed via structural differences are related to the spatial variation in subtypes of receptors for which high resolution PET atlases are available (dopamine and serotonin). 5HT2A receptor loss did appear to correlate with VH which fits with some of the current pharmacological agents used to treat VH in PD.

The study is important and adds to knowledge about possible pathophysiology of VH.

The conclusions appear robust and the discussion covers shortcomings of the data. However, also worth emphasising one challenge with studying VH in PD, is that the patient is usually not actively hallucinating in the scanner – and rather what is measured is possibly / inferred to be linked to VH.

Proof read eg line 1 ‘Patients with Parkinson’s disease (PD), aside the typical motor symptoms’

The manuscript has been amended accordingly.

We have also added the fact that we can not directly measure VH as a limitation in the discussion. Indeed, we can only show link with the trait of VH, and not the state of VH (p17).

Line 6: ‘....progress to formed hallucinations (initially with insight preserved), then hallucinations in other modalities and delusions’. I disagree that the modality of the hallucination changes – as the authors seem to have implied. I have never known a PD patient to switch from one sense to another

Thank you for your comment. We have slightly rephrased and toned down the sentence, as we meant to note that some patients may start having multimodal hallucinations, and not just visual ones, as the disease progresses, as reported for instance in ffytche et al., 2017.

The rephrased sentence now reads “There is a continuum of experiences typically characterising PDP with patients initially experiencing minor hallucinations (perception of presence or passage) and illusions that progress to formed hallucinations (initially with insight preserved); in rare cases, patients may also experience multimodal hallucinations and delusions (ffytche et al. 2017).”

Minor point – maybe shorten the manuscript

We have shortened the manuscript in several places. However, we have also made additions to address all of the reviewers’ points and so the manuscript is not shorter overall.

Reviewer #2 (Remarks to the Author)

This study deals with the structural and neuroreceptor correlates of Parkinson’s disease psychosis, mainly visual hallucinations. Authors claim that previous imaging studies investigating the neural correlates of visual hallucinations (VH) in PD have been relatively heterogeneous in their findings due to differences in study design and limitations of scale. Then, to overcome these limitations, the authors use an interesting statistical approach (empirical Bayes harmonisation) to pool together structural imaging data from multiple research groups into a large-scale mega-analysis to identify cortical regions and networks involved in VH and their relation to receptor binding. Using this methodological approach to study cortical thickness the authors claim to have found a wider cortical involvement underlying VH than previously recognised, including primary visual cortex and its surrounds, and the hippocampus, this latter being a structure that does not currently play a central role in models of VH. By interpreting the results of the structural covariance analyses they suggest a strong involvement of the attention control networks in VH. Associations found between serotonergic and dopaminergic receptor binding and cortical thickness move the authors to claim that the present study provides the first evidence that the cortical changes may be driven by neurotransmitter reductions. If confirmed, this latter finding raises the possibility of novel interventions to compensate for the neurotransmitter loss in patients suffering VH at an earlier stage of disease.

The study is interesting and the conclusions are original and potentially of interest and influence to others in the scientific and practicing community.

My main doubts and concerns are methodological and can probably be addressed by the authors:

1. I could not find any reference on how did the authors handle partial volume correction when working with PET data. That is, PET uptake should always be regionally corrected for partial volume effects, i.e for possibly-reduced gray matter density. In other words, finding a direct correlation between cholinergic/serotonergic PET uptake in a region with its cortical thickness or volume in the same region might simply be a consequence of reduced neural density in that region (and therefore lower PET uptake). This would not imply that atrophy in that region is caused by a cholinergic/serotonergic deficit, as neural loss could have been caused by another pathological process and these correlations would still hold due to partial volume effects. Therefore, caution must be taken when concluding about this type of associations.

The PET data is taken from standardised PET templates to represent the premorbid distribution of receptor densities. Therefore, areas of volumetric loss would not influence the PET uptake data. Unfortunately, there is no dataset of this size with combined MRI and PET data for Parkinson's with visual hallucinations, and so we have developed and used this method to perform neuropharmacologically based associations on structural data, in a similar way to what we have previously published in our drug studies.

We have now more clearly stressed that this is not individual PET data in the appropriate section in the Methods and at the beginning of Results paragraph 2.4.

2. The authors state that "Results were false discovery rate corrected", but do not specify the corrected p-threshold. Some results are reported at $p_{\text{fdr}} < 0.09$, which may not reach significance.

This has now been specified, and the .09 result has been removed from the text.

3. How did the authors corrected for the fact that multiple graph metrics were considered (vulnerability, transitivity, local and nodal efficiency, path length, betweenness centrality, eccentricity, distance) and correlated with multiple structural metrics (surface area, thickness...) in multiple samples?

In this setting, mild correlations reported as significant such as the one between surface area and local efficiency ($r=0.24$, $p=0.02$) could be questioned.

Permutations ($N= 5000$) were used to compute the different graph metrics. Then based on this we selected the graph metrics of interest (local and nodal efficiency) for each region at a specific density (the one whereby the greater modularity difference was observed) for PD-VH and PD-noVH and created an array of vectors, each characterised by the difference at the metric between PD-VH and PD-noVH. The relevant vector (e.g. local efficiency) was then correlated with the difference in surface area between PD-VH and PD-noVH in separate correlation analyses first and then regression models with bootstrapping. We did not analyse all the different metrics as it was not the purpose, and the text has been now amended to make this clearer. We have analysed efficiency local and nodal and have corrected p values for multiple comparisons for the measures used (local efficiency, nodal efficiency, difference of morphometrics in the full sample, in the NPI sample and in the covariance sample).

4. The authors state that "The hippocampus does not currently play a central role in models of VH in PD and our findings suggest this needs to be reconsidered". As they have already ran FreeSurfer into the T1-MRI scans, the authors could have also included in the analysis the anatomical parcellation of the hippocampus that FreeSurfer provides (<https://surfer.nmr.mgh.harvard.edu/fswiki/HippocampalSubfieldsAndNucleiOfAmygdala>). This would have allowed improved anatomical resolution in this important region.

We fully agree with the reviewer that the anatomical parcellation of hippocampus and amygdala are important analyses. Given the current focus of the paper on the cortical regions and correlates with PET maps and graph theory analyses we decided to present only the total volumes for the subcortical regions and the subdivision parcellation is the subject of a separate paper we are currently preparing, which also includes additional analyses. We have toned down our conclusion based on the fact that we have only total hippocampal volume.

We have added in the discussion: "Our results may suggest a role for the hippocampus in models of VH in PD, although detailed analysis of hippocampal subdivisions is required before this can be substantiated. We are currently exploring this issue in a separate analysis."

5. Other works in the field, albeit using VBM-GM instead of surface-based techniques, report cortical alterations (even in minor hallucinations) that are congruent with the overall results of this work. I suggest commenting them: doi:

10.1002/mds.27557, <https://doi.org/10.1111/ene.14576>, <https://doi.org/10.1101/2020.05.11.054619>

We are aware of this study and it's very interesting that there is a consistency in the results, adding robustness to the findings of both studies. We have cited the study in the discussion (p12):

“The regions of reduced thickness encompassed all cortical regions identified in previous structural imaging studies (for a review, Lenka et al., 2015), suggesting previous variability may relate to stochastic effects introduced by relatively smaller samples and design differences. Interestingly, there is substantial overlap between the regions here identified and a functional network recently found involved in presence hallucinations in both healthy and PD participants (Bernasconi et al., 2021).

With the larger sample of the mega-analysis, the extent of cortical regions involved appears wider than previously suspected.”

Reviewer #3 (Remarks to the Author)

The authors should be congratulated on the efforts of assembling the clinical and morphometric imaging data from 8 studies of Parkinson's patients with and without visual hallucination. Compared with a meta-analysis based on individual small studies and inconsistent findings, the final dataset of 493 subjects would allow enhanced statistical powers to identify group differences between PD-VH and PD-noVH. Statistical harmonization using ComBat was performed to remove site differences on the cortical thickness and surface data from 68 vertices of the Desikan-Killiany atlas. Statistical analyses were carried to compare group differences in mean and covariance of the morphometric features, and to correlate with clinical scores for visual hallucination and receptor binding. The paper is overall nicely written but might benefit from some more clarifications and details. In particular, I have the following comments:

1. Clinical characteristics: Table 1 provided p values testing PD-VH vs. PD-noVH within each study. It would also be helpful to compare the distributions of the clinical characteristics across 8 studies using e.g. boxplots. In addition, since gender was adjusted as one of the covariate, I suggest the authors to report the proportion of female as a separate column and test for difference using e.g. Chi-square test.

We have added a column for gender with the requested details in Table 1. For the distributions we provide the forest plots and the meta-analysis results, which are standardised to allow the reader to compare the distributions of clinical characteristics across the eight studies, in the supplementary materials S2.

2. Harmonization: since each study contain both VH and no-VH participants, was site differences only estimated using the no-VH patients, or was VH diagnosis adjusted in the ComBat model. In addition, is there any need for harmonization on the clinical outcome such as NPI?

We used VH / NO VH binary variable in the harmonisation model, as one would do if groups were a group of interest and a control group (besides inserting also age and gender and scanner/site). Using continuous NPI data for this kind of model would be problematic for two main reasons: first, we do not have this value for the whole sample, second, as it is a score derived from a questionnaire indicating different frequencies and intensities of hallucination symptoms, it might result in a confound in the harmonisation model.

3. For group comparison using ANOVA, Bonferroni correction was used for multiple comparison. Was each corrected for 68 regions in each analysis? Bonferroni seems could be quite conservative given the number of regions to be corrected.

As described earlier, we have modified this analysis to have more transparent multiple comparisons correction approach following the reviewer's suggestion. Initially for the MANCOVAs we had used a two-stage approach: a feature selection one-way ANOVA, and a MANCOVA using the regions screened with the ANOVA. Multiple comparisons correction was applied on the MANCOVA results. We have now decided to use a more conservative approach: we have applied multiple comparisons correction (FDR) after the

feature selection stage with the 148 regions, and only the regions surviving this correction were then entered in the MANCOVAs. This gives a slightly reduced set of regions for cortical thickness, thus we have repeated the PCA, the receptors regression models and the sensitivity analysis on the NPI subsample, all of which are largely unchanged. In addition, the results for surface area are reduced to one region, and thus surface area has been removed from PCA and receptor analyses.

Pairwise comparisons between VH and noVH were always Bonferroni corrected.

4. ANOVA and Figure 1, rather than only display p values on the brain regions, it would be more informative to indicate the effect size.

We have created a new Figure 1, with regions colour coded by effect size.

Similarly, it would be interesting to examine the direction of age association and effect sizes as well. This might be achieved by running a linear regression adjusting for age, gender and ITV.

The focus of this manuscript is on the difference between VH and noVH groups. Given the number of analyses presented and the request from other reviewers to shorten the paper as well as the extensive literature on the relationship between brain morphometrics and age and gender we decided not to include this. It may be interesting to examine the association between age, gender and ITV differences between the groups, but we feel as though this does not directly address the hypotheses of the paper. Importantly we have included age, gender and ITV in the models and the effects of these factors have been reported in the main text.

5. Receptor density analysis: I'm confused as why the estimated mean difference of the morphometrical features was used as dependent variable. Were there only one set of receptor density value per group? If I understand correctly, each dot on Figure 2 b) corresponds to a region. If so, are the magnitude of the mean difference across cortical regions comparable? Do they need to be normalized to obtain meaningful interpretation of the correlation between mean difference cortical thickness and receptor density?

To answer the first question, yes there was only one set of receptor density value for the whole sample. The PET data is taken from standardised PET templates to represent the premorbid distribution of receptor densities. There is no dataset of this size with combined MRI and PET data and we have developed and used this method to perform neuropharmacologically based analyses on volumetric data.

To address the robustness of this analysis, we have also investigated the correlations between the maps and the differences in thickness in each region taking into account spatial autocorrelation with the method used in Vasa et al., 2018 and reviewed in Markello and Misic, 2021. This is reported in the text in the receptor density maps analysis section and in the Methods.

6. PCA analysis: unless sparse PCA is performed, it is not clear how representative those claimed 'best representative' regions are compared to rest of the regions and what are the directions of association. In fact, it will be helpful to plot the contributing weights of PC 1&2 from each of the regions, or alternatively conducting post-hoc analyses to correlate the identified regions with PC scores or NPI.

We have amended the manuscript, avoiding to use 'best representative' and replaced it with 'principal components'. We have also created a plot with the contribution of the different PCs for Dimensions 1 and Dimension 2. In addition, tables with such values are available in the open science framework page of the study in the 'add-ons' section linked at the beginning of the Methods .

7. The conclusions of significance correlation between PC loadings and NPI is quite strong given that p values are quite marginal ($p=0.046$ and 0.049) and not sure if multiple comparison correction was done here.

We have rerun the PCA after changing the approach for the initial group level analysis as outlined before and we have left out that analysis.

8. Network analysis: how was cell-by-cell comparison carried out since there is only one estimated structural covariance matrix for PD-VH and PD-noVH?

We used a standard method for comparing correlation coefficients in independent cohorts, implemented in cocor in r. Cocor allows to compare correlation coefficients for two independent groups, given the coefficients and the sample size, and produces a p value and a z score. In addition, we have now corrected the p values resulting from this analysis for multiple comparisons with the Benjamini-Hochberg method and reported in this in the main text in the appropriate section.

How robust was the conclusion under different the choice of graph density?

We explored several densities as advised in the literature, and for representation and some of the analyses (correlations between efficiency metrics and thickness/area difference) we selected the density at which modularity differed the most (13%). For all the other processes several densities were explored from 5% to 20%. The results are thus robust, as when selecting the hubs, we have used degree and betweenness centrality at every density considered to isolate the brain regions acting as hubs, and chose as such only regions that respected these criteria in at least half of the densities.

Minor:

1. Would suggest to use the term 'negative correlation' rather than 'Inverse correlation'

This has been amended in the text.

Reviewer #4 (Remarks to the Author):

The authors combined several existing and new datasets to attempt to map the pattern of brain atrophy associated with visual hallucinations in Parkinson's Disease (PD). All included studies had the same goal but suffered from small sample sizes, making the current mega-analysis pertinent.

A key question is whether the theory, as assumed in some of the literature, that psychosis and hallucinations in PD are simply a result of widespread cortical involvement plus the use of dopaminergic medications is correct. An alternate theory would be that hallucinations are the

result of involvement of specific cortical areas and/or intrinsic networks. To this end they compared structural T1-weighted MRI scans of patients with and without visual hallucinations (PD-VH vs PD-noVH) using Freesurfer to measure cortical thickness, cortical surface area, and subcortical brain volumes. They used Bayesian harmonization to correct for site. The final sample was 135 PD-CH and 358 PD-noVH.

The main findings are as follows: widespread lower cortical thickness (CT) in the VH group. This seemed to encompass almost the entire cortex, with a few exceptions; lower volume of the amygdala in the VH group; some correlations between lower CT and a Neuropsychiatric Inventory of hallucinations in a subgroup. There is also an attempt to compare the cortical areas affected to maps of dopamine and 5HT receptor distributions in healthy subjects. Here it seems there was no relationship for 5HT when considering the entire cortex, but an inverse relationship if only cortical areas showing an VH-noVH effect are considered. This implies that higher 5HT innervated areas showed greater relative atrophy in the VH group. Findings for dopamine receptors were not significant. Finally, additional analyses with PCA and structural covariance more or less confirmed the initial cortical thickness analysis.

The main strength of this work is the collating of a very large dataset from multiple sites, processing the data using the same tools in one center, trying to match for age and disease stage, and using rigorous techniques to account for site effects.

However, I have a few concerns regarding the authors' interpretations and a significant concern regarding the significance of the result. Specifically I do not see how they have achieved their stated goal of: "... [disentangling] brain changes related specifically to VH mechanisms as distinct from those related to cognitive decline, PD stage or medication effects".

The most significant concern is that it is not clear that the groups are adequately matched – although the authors did make every effort to check and account for any discrepancies. In a PD sample, two factors will overwhelmingly determine cortical thickness and subcortical volumes: age and disease stage. It is difficult to tell, as some of the information is in the supplementary materials, but it seems that the VH group is older and at a more advanced stage of the disease based on UPDRS and MMSE. I am trying to ascertain if there are differences based on Table 1 and Fig. S2, but these do not appear to always match up. Nonetheless it would seem that the VH group was older and had worse MMSE (indicative of

worse disease). For UPDRS, it is not clear as the forest plot in S2 shows a negative total value, but contrasting it to Table 1 would indicate higher UPDRS scores in the VH group? In any case this part of the results needs to be clearer, and placed in the main text as it is crucial.

1)The reviewer is correct in that some details in Table 1 and Fig. S2 did not match up. The reason for this is because we do not have all data on all patients from all of the groups and our solution was to initially use as much raw data as possible, but in the groups where missing individual data was an issue we used the data as reported in the original paper. However, in some studies we had to remove participants that did not meet the study entry and quality criteria and in some cases we were provided with additional unpublished data. This means our participants pooled together are not simply the sum of the different studies and this resulted in the differences highlighted by the reviewer. We have now refined our approach.

First, Table 1 now reports when possible the values as computed on our raw data. When this was not possible due to lack of data, an asterisk * has been added in the table and a caption reading “for this variable we did not have raw data and we report the values provided in the original paper” (this is now true only for one study).

Second, we re-ran the meta-analysis with the same approach in mind: rather than running this as a summary of the published data, we used the raw data provided and when we had missing values, we excluded the participants. We think that this is the cleanest approach, however we had to remove >60 data points in order to do this (details in S2). Results do show a difference in UPDRS-III (0.37) and in MMSE (0.33) (effect size reported).

The values now reported in Table1 are exactly the same as those used as inputs for the meta-analysis in figure s2 and we have now made it clear in the table legend and in S2.

We have updated the text accordingly and updated the main case control comparison and the sensitivity analysis performed on a subsample of participants for whom we have NPI values. For the same subsample as previously reported in the sensitivity analysis, who did not differ in terms of age, gender, onset, MMSE, LEV and TIV, we added the UPDRS-III scores (except for a small number of missing values where replaced them with the mean of that study sample), MMSE and LED. The results are very similar to what was previously reported

and we have updated the text and figure accordingly (Main text 2.3, p6-7 and Supplemental S4).

For the main case control analysis (N=493) we have also analysed the subsample for whom we had MMSE and UPDRS-III data (N=440). Here we see the same regions differing between the groups for cortical thickness and surface area and no main effect of those variables. In the main text we comment on this analysis, the details of which are in the supplemental information (S3b).

2) Findings for dopamine receptors were significant, both for the all regions approach and for the VH-noVH difference and for the regions where we found no difference between the groups. The results are now more clearly described in the text (p7-8), and we have added a supplemental and methods section taking into account spatial autocorrelation due to neighbouring regions similarity for this analysis as is now recommended in the literature.

3) "... [disentangling] brain changes related specifically to VH mechanisms as distinct from those related to cognitive decline, PD stage or medication effects" refers to the results obtained in the analysis on the smaller subsample with the NPI, onset, LED, UPDRS and MMSE data, besides age, gender and TIV data. While the results support the presence of hallucinations having a distinct effect on brain morphometrics, we agree with the reviewer about our terminology here and we have toned down the language replacing 'disentangling' with 'controlling'. We have also changed 'matched' with 'do not differ' when it was the case. As we describe above the groups do display small differences in some variables for the main analysis (e.g. MMSE, UPDRS-III): for MMSE difference is 1 point (sd 0.1; with 29.4 for noVH and 28.8 for VH, both well above the cut-off not only for dementia but for cognitive impairment) and UPDRS-III difference is 5.8 (sd 2.7), with both groups being within the 'moderate' category as reported in Martínez-Martín et al., 2014. Nevertheless, we have attempted to address these in two ways. First by including these variables in an additional case control analysis with a smaller group (N=440) and devoted greater attention to the NPI subgroup analysis also in the main text.

Finally, it says in the supplementary materials that “we did not have raw data for all groups for all relevant variable...”, which makes it difficult to see whether or not the groups were truly matched. Obviously, a lack of a $p < 0.05$ difference does not mean a lack of important effect. Also it is not necessarily the case that putting age as a confound will eliminate the bias.

Yes, we agree that a lack of $p < 0.05$ difference doesn't mean that the groups are matched. We have changed the terminology and now say ‘we do not have evidence that the group differed’ for instances where $p > 0.05$. We also agree that these factors can theoretically influence the results and include additional analyses either adding these factors as covariates, or focussing only on the PD-VH group (see above). For age the mean difference was only 2.2 years and while this was included in the models it is extremely unlikely to explain the findings based on the narrow effects of age from a multitude of published studies.

Studies have shown that when comparing longitudinally individuals after 2 years the occipital cortex is least affected in brain aging whereas the hippocampus is the most affected region (see Peters 2006, see also Raz et al., 2004). Interestingly, the hippocampus was observed as reduced in volume in VH participants in our subsample where participants not only did not differ in terms of age, but were also controlled for all the other relevant variables, besides having all those variables and the relevant interaction terms as covariates. Indeed, atrophy in the hippocampus increases after 70 years old (Scahill et al., 2003). Both our groups mean age is below 70 with individuals being both well above and below the mean, but none in very old age (PD-VH = 67.85, SD = 7.74; and PD-noVH = 65.66, SD = 8.71).

Secondly, we have conducted a subgroup analysis for those participants for which we had more complete data available, including the NPI. For this sensitivity analysis there were 146 participants who did not differ for age, gender, onset, LED, MMSE and TIV. For the UPDRS-III it was more complicated because we had some missing data even in this smaller sample. The results of this analysis (reported in S4 and mentioned in the main text) largely overlap with those in the main sample. In addition, we have added for this revision a “leave one group out” type of analysis, comparing VH and noVH patients in all groups minus one, rotating the excluded group in order to cover the whole sample, obtaining consistent results.

Many in the dementia field are now using the W-score approach to account for age and sex

effects on MRI measures – as outlined in Journal of Neuroscience 32(46):16265-16273, but this would require healthy controls scanned at the same sites.

Unfortunately, healthy controls data for all sites are not available. If healthy controls data would have been accessible for all sites, we would have used a normative modelling approach to account for this.

Questions regarding the Interpretations:

1. It is not clear what the main conclusion of the study is. If it is that VH is associated with greater cortical loss, this is not surprising. As mentioned by the authors, psychosis and hallucinations are a harbinger of dementia and loss of autonomy, likely indicating widespread pathology. There is an extensive postmortem literature on visual hallucinations in PD, going back decades, which consistently associates this symptom with greater Lewy pathology in limbic and cortical areas.

We extend the literature by demonstrating these effects in a large international cohort looking at all brain regions and further allowing us to go beyond localised univariate analysis and look at patterns across the brain in relation to i) receptors, ii) interrelationships between areas. We have added more on the postmortem literature in the discussion and on how our study complements it (p14).

2. The PCA analysis does not seem to add anything. It seems a more complicated analytical approach that also concludes in reduced cortical thickness in the VH group. A similar argument can be made for the structural covariance analysis. In this analysis only two thresholded correlation matrices (one per group) are compared with a wide array of network measures. The conclusion that the group differences especially involve the dorsal and ventral attention networks is difficult to follow since it is not supported by the more straightforward original group differences analysis. A systematic analysis of cortical thickness differences for each intrinsic network might better prove this point.

The PCA formally acknowledges that the regions are unlikely to be independent and is a standard data reduction approach which accommodates this covariance structure.

The discussion of the paper states that the differences in cortical thickness encompassed almost all brain areas listed, and the PCA demonstrates that there are organisational principles across the brain in these differences with two statistically separable dimensions. However, the conclusion about the dorsal and attentional network involvement comes from the separate network analysis. We have reviewed the discussion to ensure we clearly state that the conclusions about this are derived from the network analysis. We have also used, from the covariance analysis sample, the regions considered to be part of the DAN, VAN and DMN as advised in this second remark about the interpretation: we have compared the means of thickness and area in such regions in noVH vs VH (with age and gender as covariates). For this purpose we have collated together the mean of the regions making up each network and compared patients on those. For thickness we find significant differences in the VAN and the DAN, and only a trend for the DMN, whereas we do not find any difference for surface area. This result is consistent with our suggestion that the group differences particularly involve the dorsal and ventral attention networks. In addition, both results are consistent with the differences found in the single regions making up such network in the main sample analysis. These are the regions we used, based on Shine et al., 2014, 2015.

DAN

Dorsolateral PFC – IFG opercularis and triangularis

Posterior parietal cortex - supramarginal gyrus

Frontal eye fields – precentral g

Corpus striatum – this was not available from the current analyses, but will be taken into account in the subcortical subfields study we are preparing

VAN

Basolateral amygdala - amygdala – this was not included in this analysis as the measurement is different for subcortical volumes than for thickness and area. However differences in the amygdala were reported here for both the main and the NPI sample

Lateral and inferior PFC – IFG orbitalis, lateral orbitofrontal cortex (IOFC)

Temporoparietal junction - used the inferior parietal lobule (IPL)

Ventral striatum – as for the amygdala, we did not include the accumbens in this analysis, but we had it in the main analysis and no difference was observed in this region.

DMN

MT – PHG and enthorinal

Medial PFC - medial orbitofrontal cortex (mOFC) and frontopolar region

Posterior cingulate cortex

Nevertheless, this analysis mainly summarises structural differences possibly related to neural pathology in these networks, whereas the findings of the structural covariance analysis investigate network-level properties related to cortical thinning and anomalies in surface area taking into account not only morphometric differences but connectivity metrics.

This is now reported in Supplementary Information **SI8b**.

3. The conclusion that degeneration in 5HT neurons may lead to cortical degeneration in projection sites is not at all supported by this study – as this was not the point of the experiment. An analysis with control subjects' MRI data at the minimum would be needed to assert that 5HT projection sites are more vulnerable in PD. But even then, this would not implicate loss of 5HT projections in cortical neurodegeneration, as 5HT innervation was not measured in this study. Finally, this disregards the evidence that PD is caused by synucleinopathy that affects many neuronal types.

The reviewer is correct in that this study cannot demonstrate 5-HT neuronal loss leads to cortical degeneration. Our comment here was intended as hypothesis generating and longitudinal studies will be required to demonstrate this. This has now been also stressed in the discussion. Specifically, the neuroreceptors maps used in the study are derived from independent healthy participants PET scans thus providing a good representation of pre-morbid receptor distributions. Thus, using this data as predictor of morphometric differences in the two groups, we can infer which premorbid receptor distribution may be most affected in VH in PD. We find that all receptors considered (DA, 5-HT) seem to be involved, supporting the multiple neural types affected by synucleinopathy in PD. We agree that future work should include also other receptor types and we are currently gathering data to do so. This is a prediction from this data but longitudinal studies will be required to demonstrate this in the future as we stress on p18.

Minor concerns:

1. The authors state that there is only “sparse Lewy pathology in the cortex of PD patients with VH at the stage included in our analysis”. The citation to Harding et al. Brain 2002 is inappropriate as this is a study of amygdala pathology only. Moreover, there have been many postmortem studies showing diffuse cortical Lewy pathology in patients with hallucinations.

We cite the study by Harding and colleagues as it found that PD patients with VH and MMSE score >25, as it is the case for our study, there was indeed an increase in Lewy body pathology in the basolateral nucleus of the amygdala, however they note Lewy bodies were only sparsely present in the cortex and in the hippocampus. This is interesting for us as in the subsample analysis when adding MMSE as a covariate we found a difference in the hippocampus that was not found for the main sample, when the MMSE score was not available. The relative absence of cortical or hippocampal pathology makes it more likely that visual hallucinations in these patients depend upon wider functional changes in brain networks and that this is related to neurotransmitters, rather than reflecting localized neuropathology. We agree with the reviewer that postmortem studies show diffuse Lewy pathology as well as Alzheimer’s pathology but this evidence is derived from end stage

disease, long after the development of VH, with no evidence to suggest the neuropathological changes are present to the same extent at the disease stage studied here.

2. I could not tell if UPDRS-III was on or off medications.

We have ascertained that UPDRS-III scores were collected on medication.

REVIEWERS' COMMENTS

Reviewer #1 (Remarks to the Author):

The authors have revised according to reviewers comments. no further comments. The article provides new information regarding potential cortical regions involved in VH in PD.

Reviewer #2 (Remarks to the Author):

The authors have worked hard to try to address the major and minor concerns raised by this reviewer. Most of the methodological items have been satisfactorily addressed. In my opinion the paper has substantially improved and, in its present version may contribute interesting new data to the field.

Reviewer #3 (Remarks to the Author):

The authors have addressed most of my questions in the revised manuscript and through extra analyses. I have no additional comments.

Reviewer #4 (Remarks to the Author):

The authors have revised the paper in response to my and other reviewers' comments. The main strength of the paper remains an unprecedented sample size to map cortical involvement in visual hallucinations in PD. I agree with all but one of the rebuttals to my initial review, as follows:

Regarding this statement: "An earlier study of PD patients with visual hallucinations and MMSE>25 (similar to our cohort) found only sparse Lewy body pathology in the cortex of PD patients with VH at the disease stage included in our analysis (Harding et al., 2002), raising the question of what causes the extensive cortical changes found in this and previous studies".

I re-read the Harding et al. paper and the only mention of cortex is this sentence: "None of the selected Parkinson's disease cases had more than isolated cortical LB in the cingulate, hippocampal or association cortices." There is no data presented on these regions nor a reference to another paper. Also, this says nothing about other pathology besides Lewy Bodies. I am not sure this throw-away sentence allows one to conclude that the patients in the current meta-analysis lack cortical Lewy pathology.

Also, I still don't see the logic of the authors' argument. The cortical atrophy is presumably caused by PD, whether or not Lewy bodies are abundant. I certainly agree that cortical thinning likely represents synaptic loss, which could include monoaminergic projections, but could also include other synapses, and might occur in the absence of extensive Lewy bodies in the cortex since these are in the cell bodies. If the cortical thinning is due to loss of synapses – that is still neuropathology.

In any case the statement in the summary that neurotransmitter loss must drive cortical thinning is not supported by these data. Showing an overlap between the atrophy and PET maps does not prove causality. Similarly I don't see the justification for "Finally, the findings shed light on why widespread cortical changes occur at a stage of PD with only sparse cortical neuropathology".

But I note that this is not the main point of the paper.

More generally, I do not see a clear argument based on these correlational results for the role of loss of cortical serotonin and dopamine signaling in the mechanism of visual hallucinations.

--

Bernasconi et al. is not in the reference list.

Reviewer #1 (Remarks to the Author):

The authors have revised according to reviewers comments. no further comments. The article provides new information regarding potential cortical regions involved in VH in PD.

We wish to thank Reviewer #1 for the time spent on our manuscript and their comments and suggestions.

Reviewer #2 (Remarks to the Author):

The authors have worked hard to try to address the major and minor concerns raised by this reviewer. Most of the methodological items have been satisfactorily addressed. In my opinion the paper has substantially improved and, in its present version may contribute interesting new data to the field.

We wish to thank Reviewer #2 for the time spent on our manuscript and their comments and suggestions.

Reviewer #3 (Remarks to the Author):

The authors have addressed most of my questions in the revised manuscript and through extra analyses. I have no additional comments.

We wish to thank Reviewer #3 for the time spent on our manuscript and their comments and suggestions.

Reviewer #4 (Remarks to the Author):

The authors have revised the paper in response to my and other reviewers' comments. The main strength of the paper remains an unprecedented sample size to map cortical involvement in visual hallucinations in PD. I agree with all but one of the rebuttals to my initial review, as follows:

Regarding this statement: "An earlier study of PD patients with visual hallucinations and MMSE>25 (similar to our cohort) found only sparse Lewy body pathology in the cortex of PD patients with VH at the disease stage included in our analysis (Harding et al., 2002), raising the question of what causes the extensive cortical changes found in this and previous studies".

I re-read the Harding et al. paper and the only mention of cortex is this sentence: "None of the selected Parkinson's disease cases had more than isolated cortical LB in the cingulate, hippocampal or association cortices." There is no data presented on these regions nor a reference to another paper. Also, this says nothing about other pathology besides Lewy Bodies. I am not sure this throw-away sentence allows one to conclude that the patients in the current meta-analysis lack cortical Lewy pathology.

Also, I still don't see the logic of the authors' argument. The cortical atrophy is presumably caused by PD, whether or not Lewy bodies are abundant. I certainly agree that cortical thinning likely represents synaptic loss, which could include monoaminergic projections, but

could also include other synapses, and might occur in the absence of extensive Lewy bodies in the cortex since these are in the cell bodies. If the cortical thinning is due to loss of synapses – that is still neuropathology.

In any case the statement in the summary that neurotransmitter loss must drive cortical thinning is not supported by these data. Showing an overlap between the atrophy and PET maps does not prove causality. Similarly I don't see the justification for "Finally, the findings shed light on why widespread cortical changes occur at a stage of PD with only sparse cortical neuropathology".

But I note that this is not the main point of the paper.

More generally, I do not see a clear argument based on these correlational results for the role of loss of cortical serotonin and dopamine signaling in the mechanism of visual hallucinations.

We wish to thank Reviewer #4 for the time spent on our manuscript and their comments and suggestions. With regards to the remaining concerns expressed by Reviewer #4 they have very helpfully drawn attention to our use of the term 'neuropathological' and the possibility of it being misconstrued by the readership. In fact, the point we were hoping to make is the same as the reviewer's and we have reworded the relevant sections to resolve any ambiguity. With regards to the sentence the reviewer quotes and there is no justification for, we have changed the sentence.

Bernasconi et al. is not in the reference list.

Thank you, this has been amended.